# IMPROVED GENERALIZATION OF CGAN USING VICINAL ESTIMATION

## ABSTRACT

The problem of generating high-dimensional distributions has been known as a difficult problem in machine learning due to the *Curse of Dimensionality*: the higher the dimensionality is the more the empirical data deviates from its original distribution even for a large number of samples. Along with the Curse of Dimensionality, the generalization of **conditional density estimation (CDE)** suffers from so-called *Lack of Conditional Samples*: the number of data for each conditional density is usually much smaller than the number of samples or no data is avaiable for some conditional densities. To overcome these difficulties, we introduce the concept of **Vicinal Estimation (VE)** which is shown to be useful in estimating conditional densities. With VE we propose a **conditional Generative Adversarial Network (cGAN)** model and analyze theoretically that the generalization error of our model is independent of the dimensionality of the output. We also show that our theoretical analysis holds in practice through experiments.

## 1 INTRODUCTION

For many years **Generative Adversarial Networks (GANs)** (Goodfellow et al., 2014) have been widely used to predict high dimensional distributions like **images**. Despite their widespread use and the existence of models that perform well on high-resolution images (Brock et al., 2019; Karras et al., 2019), it is still an important issue that how we can stablize the training of GANs and make them have good generalization performance. GANs are usually trained to minimize the distance between the generated distribution and the empirical distribution, but the estimation of the empirical distribution is prone to failure especially in high dimension. This phenomenon is called the *Curse of Dimensionality* and it is the main reason why the generalizaton error analysis of GANs is very difficult. For example, it is known (Weed & Bach, 2017) that for every $d$ and $\delta > 0$, there exists $C_{d,\delta} > 0$ such that the Wasserstein-2 distance between the empirical distribution $p_X$ and its true distribution $p$ satisfies the inequality

$$\mathbb{E}_X[W_2(p_X, p)] \geq C_{d,\delta} N^{-\frac{1}{d-\delta}} \tag{1}$$

where $X$ denotes an i.i.d. sampling of size $N$ from the true distribution $p$ on $[0,1]^d$ which is absolutely continuous with respect to the Lebesgue measure.

The curse of dimensionality issue is exacerbated when predicting *conditional* distributions with **conditional GANs (cGANs)** (Mirza & Osindero, 2014) for **Conditional Density Estimation (CDE)**. In CDE we have to handle the issue that the training data for each conditional variable (called **label** in this work) is not sufficient or does not exist. This issue becomes even worse when the label is continuous (as opposed to class-conditional) because there are an infinite number of possible values for the label, but a finite number of training data. In this work, we call this issue *Lack of Conditional Samples*. To solve this issue, we need to consider a method to increase the *effective* number of data in CDE. All the above issues are closely related with the generalization performance of cGANs and some recent works provide partial solutions to the issues as follows:

- One recent work (Yang & E, 2021) introduces a simplied GAN model in unconditional density estimation using early stopping and adding a regularization term based on RKHS norm and it can achieve a dimension-free Wasserstein-2 generalization error, which allows the GAN model to escape from the *Curse of Dimensionality*. However, the same approach cannot be applied to cGANs because we still have the *Lack of Conditional Samples* issue.

- Another recent work called **Continuous cGAN (CcGAN)** (Ding et al., 2023) considers a training loss of cGAN based on **Vicinal Risk Minimization (VRM)** (Chapelle et al., 2000). This approach augments the label with a perturbation to increase the effective number of data in CDE, so it partially solves the *Lack of Conditional Samples* issue. However, (Ding et al., 2023) only analyzes the error bound on the discriminator, not on the generated conditional density. Furthermore, it does not handle the *Curse of Dimensionality* issue and only focuses on one-dimensional label.

In this work, we combine and further extend the ideas of the above two works to develop a new cGAN model which provides good generalization performance as shown later even in high dimension. In our cGAN model we introduce the concept of **Vicinal Estimate (VE)** of a distribution, which is an alternative distribution of an **auxiliary label** value which replaces the original label value. In this case, the conditional density of VE becomes a linear combination of original conditional densities, which can resolve the *Lack of Conditional Samples* issue. We then propose our cGAN model to predict the conditional density of the VE. We analyze it using a similar approach as given in (Yang & E, 2021) to obtain a generalization error bound on VE independent of the output dimensionality. Finally, we complete our analysis on generalization error by converting the bound on VE to that on the original distribution and selecting an optimal auxiliary label distribution.

The remainder of this paper is organized as follows: We review related works in section 2 and introduce the notation, the problem setup, and the basic cGAN framework in section 3. In section 4, we define the VE of a general distribution and derive its properties including the conditional densities of the VE and their differences from the original conditional distributions. We also present our cGAN model for estimating the conditional densities of the VE. The analysis on generalization error is provided in section 5, and experimental results are provided in section 6. Finally, section 7 concludes this paper with a discussion of future works. Proofs and Experimental details are provided in appendixes.

## 2 RELATED WORKS

**Vicinal Risk Minimization:** VRM is widely used as a theoretical basis on data augmentation (Simard et al., 1998), primarily in supervised learning. For example, mixup (Zhang et al., 2018) is one of widely used data augmentation methods that is proved to improve the robustness and generalization of the models (Zhang et al., 2021). However, to the best of the authors' knowledge, our work is the first one to generalize the notion of VE and to investigate the generalization error from the CDE perspective.

**cGANs with continuous labels:** Since most of the state-of-the-art cGAN algorithms (Odena et al., 2017; Miyato & Koyama, 2018) focus on class-conditional labels, research works focusing on generating conditional distributions over continuous labels are rarely found in the literature and CcGAN (Ding et al., 2023) is known to be the first work with continuous labels. There are some research works (Jahanian et al., 2020; Shoshan et al., 2021) that control the images by changing the latent variable of an unconditional GAN according to the label value, but such methods are not suitable for generating conditional densities, which is the main interest of this paper.

**Dimension-free Generalization of GANs:** Since it is very difficult to analyze dimension-free generalization errors of GANs, the analysis is usually conducted with simplified generators (Feizi et al., 2018; Lei et al., 2020; Wu et al., 2019) or simplified discriminators (E et al., 2021; Yang & E, 2021). For cGANs, there exist no works on dimension-free generalization error, but we consider a simplified discriminator and its corresponding function space called Barron space and derive a generalization error bound that is independent of the output dimensionality.

## 3 PROBLEM SETUP

We first set up some notation and describe CDE and a cGAN framework. In this study, $\mathbf{x}$ denotes label and $\mathbf{y}$ denotes image.

Let $\mathcal{X} \subset \mathbb{R}^{d_{\boldsymbol{x}}}$ and $\mathcal{Y} \subset [0,1]^{d_{\boldsymbol{y}}}$ be the domains of all possible labels and images, respectively. We assume that there is a joint distribution $p_r(\boldsymbol{x}, \boldsymbol{y})$ on $\mathcal{X} \times \mathcal{Y}$ which we call **true distribution**. Then

**Conditional Density Estimation (CDE)** is a supervised learning task to estimate the conditional density $p_r(\boldsymbol{y}|\boldsymbol{x})$. We observe a dataset of $N$ data pairs $(X, Y) = \{(\boldsymbol{x}_i, \boldsymbol{y}_i)\}_{i=1}^N$ and aim to estimate $p_r(\boldsymbol{y}|\boldsymbol{x})$. With a given dataset $(X, Y)$, we obtain an **empirical distribution**

$$p_{X,Y}(\boldsymbol{x}, \boldsymbol{y}) = \frac{1}{N}\sum_{i=1}^N \delta(\boldsymbol{x}-\boldsymbol{x}_i)\delta(\boldsymbol{y}-\boldsymbol{y}_i) \tag{2}$$

instead of $p_r(\boldsymbol{x}, \boldsymbol{y})$. Note that, to achieve the objective of CDE, we have to focus on the distance between conditional densities, not on the joint distribution. In addition, we have to analyze the **generalization error** in CDE for a predicted distribution $p$ which is defined in this work by

$$\mathbb{E}_{\boldsymbol{x}\sim p_r(\boldsymbol{x})}L(p(\boldsymbol{y}|\boldsymbol{x}), p_r(\boldsymbol{y}|\boldsymbol{x})) \tag{3}$$

where $L(\mu, \nu)$ is any metric between two probabilistic measures. From the definition of the conditional density

$$p_r(\boldsymbol{y}, \boldsymbol{x}) = p_r(\boldsymbol{x}) \cdot p_r(\boldsymbol{y}|\boldsymbol{x}), \tag{4}$$

we can say that the procedure of CDE is twofold: (1) Estimating the marginal distribution of the label $p_r(\boldsymbol{x})$ and then (2) estimating the conditional density $p_r(\boldsymbol{y}|\boldsymbol{x})$. In this work, we assume that the label distribution $p_r(\boldsymbol{x})$ is an already known distribution on $\mathcal{X}$ to focus on the second problem only because the first problem is actually an *unconditional* density estimation on $p_r(\boldsymbol{x})$, and $p_r(\boldsymbol{x})$ in (4) can be replaced to any distribution on $\mathcal{X}$ without affecting the conditional density $p_r(\boldsymbol{y}|\boldsymbol{x})$. The other reason for this assumption is that cGAN, which we consider in this work, predicts only the conditional density $p_r(\boldsymbol{y}|\boldsymbol{x})$, not $p_r(\boldsymbol{x})$.

In cGAN, we seek to find a generator $G : \mathcal{Z} \times \mathcal{X} \to \mathcal{Y}$ to reproduce the conditional density of the training set $(X, Y)$ with respect to the label value where $\mathcal{Z}$ is a space of **latent variable** $Z$ which is a random variable with a pre-determined distribution $p_Z(\boldsymbol{z})$ (usually Gaussian). cGAN trains its generator in an adversarial manner by introducing an auxiliary network called discriminator $D : \mathcal{Y} \times \mathcal{X} \to \bar{\mathbb{R}}$ and transforming a minimization problem on conditional density into a minimax problem. After training we use the trained generator $G$ to get a conditional density $p_g(\boldsymbol{y}|\boldsymbol{x})$ derived from $G(Z, \boldsymbol{x})$ by considering $Z$ as a latent random variable where $\boldsymbol{x}$ is a fixed label value.

For example, the Wasserstein cGAN (Arjovsky et al., 2017) uses the minimax problem $\max_G \min_D V(G, D)$ with the following cGAN loss

$$V(G, D) = \mathbb{E}_{\boldsymbol{x}\sim p_r(\boldsymbol{x})}\left[\mathbb{E}_{\boldsymbol{y}\sim p_r(\boldsymbol{y}|\boldsymbol{x})}D(\boldsymbol{y}, \boldsymbol{x}) - \mathbb{E}_{\boldsymbol{z}\sim p_Z(\boldsymbol{z})}D(G(\boldsymbol{z}, \boldsymbol{x}), \boldsymbol{x})\right] \tag{5}$$

where $D$ is restricted to be a 1-Lipschitz continuous function with respect to $\mathbf{y}$. For an optimal discirminator, it is equivalent to minimizing the expected value of the Wasserstein-1 divergence between two conditional densities:

$$\mathcal{L}(p_g) = \mathbb{E}_{\boldsymbol{x}\sim p_r(\boldsymbol{x})}W_1(p_g(\boldsymbol{y}|\boldsymbol{x}), p_r(\boldsymbol{y}|\boldsymbol{x})). \tag{6}$$

which matches with a generalization error of CDE.

However, the exact value of $\mathcal{L}(p_g)$ is not tractable as we do not know the true distribution $p_r$ in general. Therefore, we instead try to calculate its **empirical risk**

$$\hat{\mathcal{L}}(p_g) = \mathbb{E}_{\boldsymbol{x}\sim p_X(\boldsymbol{x})}L(p_g(\boldsymbol{y}|\boldsymbol{x}), p_{X,Y}(\boldsymbol{y}|\boldsymbol{x})). \tag{7}$$

which is tractable. In fact, however, empirical risk is not a reasonable estimate of the generalization error because of the Curse of Dimensionality and Lack of Conditional Samples which are the issues discussed in the previous sections.

## 4 VICINAL ESTIMATION

To overcome the *Lack of Conditional Samples* issue, we introduce the **Vicinal Estimation (VE)** of a distribution. Let $\mathcal{X}' \subset \mathbb{R}^{d_{\boldsymbol{x}'}}$ be the domain of **auxiliary label** and assume that we have an **auxiliary distribution** $q_{\boldsymbol{x}}$ on $\mathcal{X}'$, which is a distribution defined for every $\boldsymbol{x}$. Once we have an auxiliary distribution, we can define the **vicinal estimate** $\tilde{p}$ of a joint distribution $p$ with $q_{\boldsymbol{x}}$:

$$\tilde{p}(\boldsymbol{x}', \boldsymbol{y}) = \int_{\mathcal{X}} p(\boldsymbol{x}, \boldsymbol{y})q_{\boldsymbol{x}}(\boldsymbol{x}')d\boldsymbol{x}. \tag{8}$$

---

**Algorithm 1** Sampling algorithm for $\tilde{p}(\boldsymbol{x}', \boldsymbol{y})$

---

**Input:** $n \in \mathbb{N}$: Number of sampled data
  $\{(\boldsymbol{x}_i, \boldsymbol{y}_i)|1 \leq i \leq n\} \leftarrow n$ sampled data from $p(\boldsymbol{x}, \boldsymbol{y})$
  **for** $1 \leq i \leq n$ **do**
    $\boldsymbol{x}_i' \leftarrow$ sample from the auxiliary distribution $q_{\boldsymbol{x}_i}(\boldsymbol{x}')$
  **end for**
**Output:** $\{(\boldsymbol{x}_i', \boldsymbol{y}_i)|1 \leq i \leq n\}$: $n$ sampled data from $\tilde{p}(\boldsymbol{x}', \boldsymbol{y})$

---

Intuitively, it is equivalent to *mapping* a true label $\boldsymbol{x}$ to an auxiliary label $\boldsymbol{x}'$ after sampling $(\boldsymbol{x}, \boldsymbol{y})$ from the original distribution $p$ by using Algorithm 1.

We can see that the mapping of label is independent of $\boldsymbol{y}$, meaning that any conditional density of the VE can be written in terms of the original conditional density:

$$\tilde{p}(\boldsymbol{y}|\boldsymbol{x}') = \frac{\tilde{p}(\boldsymbol{x}', \boldsymbol{y})}{\int_{\mathcal{Y}} \tilde{p}(\boldsymbol{x}', \boldsymbol{y}) d\boldsymbol{y}} = \frac{\int_{\mathcal{X}} p(\boldsymbol{y}|\boldsymbol{x}) p(\boldsymbol{x}) q_{\boldsymbol{x}}(\boldsymbol{x}') d\boldsymbol{x}}{\int_{\mathcal{X}} p(\boldsymbol{x}) q_{\boldsymbol{x}}(\boldsymbol{x}') d\boldsymbol{x}} = \int_{\mathcal{X}} p(\boldsymbol{y}|\boldsymbol{x}) \tilde{q}_{\boldsymbol{x}'}(\boldsymbol{x}) d\boldsymbol{x} \tag{9}$$

where

$$\tilde{q}_{\boldsymbol{x}'}(\boldsymbol{x}) = \frac{p(\boldsymbol{x}) q_{\boldsymbol{x}}(\boldsymbol{x}')}{\int_{\mathcal{X}} p(\boldsymbol{x}) q_{\boldsymbol{x}}(\boldsymbol{x}') d\boldsymbol{x}}. \tag{10}$$

So $\tilde{p}(\boldsymbol{y}|\boldsymbol{x}')$ is an expected value of the original conditional density $p(\boldsymbol{y}|\boldsymbol{x})$ under a modified distribution $\tilde{q}_{\boldsymbol{x}'}(\boldsymbol{x})$ of $\boldsymbol{x}$. Note that $\tilde{q}_{\boldsymbol{x}'}(\boldsymbol{x})$ is actually the conditional density of $\boldsymbol{x}$ given the value of $\boldsymbol{x}'$, which is exactly the *inverse* of what $q_{\boldsymbol{x}}(\boldsymbol{x}')$ describes. Hence, we call it the **inverse auxiliary distribution**.

Note that a VE reduces the conditional information of labels to a certain extent. For example, uniform auxiliary distributions on $\mathcal{X}'$ erase completely the information of labels in the data, making the VE of the data the same for all auxiliary label values. Therefore, $\tilde{q}$ needs to be *discriminative* to make the VE close enough to its original distribution. We show that the upper bound of $\tilde{q}$ defined by

$$||\tilde{q}||_{\infty} = \sup_{\boldsymbol{x} \in \mathcal{X}, \boldsymbol{x}' \in \mathcal{X}'} \tilde{q}_{\boldsymbol{x}'}(\boldsymbol{x}), \tag{11}$$

which represents the magnitude on how the inverse auxiliary distribution is distributed smoothly, plays an important role in generalization error of $\tilde{p}$, which will be clear in the next section.

Note that the original distribution can be also considered as a VE by setting $\mathcal{X}' = \mathcal{X}$ and $q_{\boldsymbol{x}}(\boldsymbol{x}') = \delta(\boldsymbol{x} - \boldsymbol{x}')$, but we immediately have $\tilde{q}_{\boldsymbol{x}'}(\boldsymbol{x}) = \delta(\boldsymbol{x} - \boldsymbol{x}')$ in this case, which means that both auxiliary distribution and inverse auxiliary distribution are not bounded.

In this paper, we focus on the **label Lipschitz conditional densities**. For a constant $L > 0$, we call the conditional density $p(\cdot|\boldsymbol{x})$ is $L$-**label Lipschitz** if $W_2(p(\cdot|\boldsymbol{x}_1), p(\cdot|\boldsymbol{x}_2)) \leq L||\boldsymbol{x}_1 - \boldsymbol{x}_2||$ for every $\boldsymbol{x}_1, \boldsymbol{x}_2 \in \mathcal{X}$. For those distributions, we can construct a bound for $\tilde{p}(\cdot|\boldsymbol{x})$ from its original distribution:

**Lemma 1.** *Suppose that $p$ is $L$-label Lipschitz. Then*

$$W_2(\tilde{p}(\cdot|\boldsymbol{x}'), p(\cdot|m_{\tilde{q}}(\boldsymbol{x}'))) \leq L\sqrt{\mathbb{E}_{\tilde{q}_{\boldsymbol{x}'}(\boldsymbol{x})}[||\boldsymbol{x} - m_{\tilde{q}}(\boldsymbol{x}')||^2]} \tag{12}$$

*where $m_{\tilde{q}}(\boldsymbol{x}') = \mathbb{E}_{\boldsymbol{x} \sim \tilde{q}_{\boldsymbol{x}'}(\boldsymbol{x})}[\boldsymbol{x}]$ is the center of mass of the distribution $\tilde{q}_{\boldsymbol{x}'}$.*

Note that the quantities $m_{\tilde{q}}(\boldsymbol{x}')$ and $d_{\tilde{q}}(\boldsymbol{x}') := \sqrt{\mathbb{E}_{\tilde{q}_{\boldsymbol{x}'}(\boldsymbol{x})}[||\boldsymbol{x} - m_{\tilde{q}}(\boldsymbol{x}')||^2]}$ only depend on the inverse auxiliary distribution $\tilde{q}$, not on the distribution $p$. In fact, $\tilde{q}$ is the main point of interest in this study as it holds all the information of vicinal estimation contributing to the generalization bound as shown later.

One simple method to construct $\tilde{q}$ is to make $\mathcal{X}' = \mathcal{X}$ and $\tilde{q}_{\boldsymbol{x}'}$ is a distribution centered at $\boldsymbol{x}'$ (i.e., $m_{\tilde{q}}(\boldsymbol{x}') = \boldsymbol{x}'$). We call such $\tilde{q}$ a **perturbation** in this paper. Here, we introduce two simple examples of $\tilde{q}$, inspired from the setting used in continuous cGAN (Ding et al., 2023).

---

**Algorithm 2** Sampling algorithm of $\tilde{p}_g(\boldsymbol{y}|\boldsymbol{x}')$ from $G$

---

**Input:** $\boldsymbol{x}' \in \mathcal{X}'$ : Auxiliary label, $n \in \mathbb{N}$ : Number of sampled data

    $\{\boldsymbol{x}_i | 1 \leq i \leq n\} \leftarrow n$ samples from $\tilde{q}_{\boldsymbol{x}'}(\boldsymbol{x})$

    $\{\boldsymbol{z}_i | 1 \leq i \leq n\} \leftarrow n$ samples from $p_Z(\boldsymbol{z})$

    **for** $1 \leq i \leq n$ **do**

        $\boldsymbol{y}_i \leftarrow G(\boldsymbol{z}_i, \boldsymbol{x}_i)$

    **end for**

**Output:** $\{\boldsymbol{y}_i | 1 \leq i \leq n\}$ : sample from $\tilde{p}(\boldsymbol{y}|\boldsymbol{x}')$

---

- $\tilde{q}_{\boldsymbol{x}'}(\boldsymbol{x})$ is a uniform distribution on the box $\boldsymbol{x}' + [-\kappa, \kappa]^{d_{\boldsymbol{x}}}$ where $\kappa > 0$ is a hyperparameter.
- $\tilde{q}_{\boldsymbol{x}'}(\boldsymbol{x})$ is a multivariate Gaussian distribution $\mathcal{N}(\boldsymbol{x}', \sigma^2 I)$ where $\sigma > 0$ is a hyperparameter.

In both cases, we can show that $d_{\tilde{q}}(\boldsymbol{x}')$ is a constant multiple of $||\tilde{q}||_\infty^{-1/d_{\boldsymbol{x}}}$. In fact, we can show the same property for a wide variety of the inverse auxiliary distributions including various perturbations. Consider a family $\{\tilde{q}^L\}$ of the inverse auxiliary distributions $\tilde{q}_{\boldsymbol{x}'}^L(\boldsymbol{x})$ parametrized by the hyperparameter named **lengthscale** $L > 0$, which satisfies the following:

- $m_{\tilde{q}^L}(\boldsymbol{x}')$ is the same for every $L > 0$.
- The random variable $(\boldsymbol{x} - m_{\tilde{q}^L}(\boldsymbol{x}'))/L$ is the same for every $L > 0$ and $\boldsymbol{x}' > 0$ where $\boldsymbol{x} \sim \tilde{q}_{\boldsymbol{x}'}(\boldsymbol{x})$.

With the above assumptions, we can observe that $d_{\tilde{q}^L}(\boldsymbol{x}') \propto L^{-1}$ and $||\tilde{q}^L||_\infty \propto L^{-d_{\boldsymbol{x}}}$. So we can show the following theorem:

**Theorem 1.** *For the family $\{\tilde{q}^L\}$, there exists a constant $C > 0$ such that $d_{\tilde{q}^L}(\boldsymbol{x}') = C||\tilde{q}^L||_\infty^{-1/d_{\boldsymbol{x}}}$ holds for every $L > 0$.*

In this paper, we assume that $\tilde{q}$ is given to have a constant $D > 0$ such that $d_{\tilde{q}}(\boldsymbol{x}') \leq D||\tilde{q}||_\infty^{-1/d_{\boldsymbol{x}}}$. In this case, the difference between $p$ and $\tilde{p}$ is bounded by the *inverse* of $||\tilde{q}||_\infty$, meaning that the *larger* $||\tilde{q}||_\infty$ the better. On the other hand, a *smaller* value of $||\tilde{q}||_\infty$ is required to have a better generalization bound on the VE as shown later. Therefore, the optimal value of $||\tilde{q}||_\infty$ should be carefully selected to make the best generalization bound on $p$, which is the objective of our analysis and will be done in section 5.

### 4.1 cGAN using Vicinal Estimate

We develop a new cGAN model using the idea of VE in this section. To use VE in our cGAN model, we train our cGAN model (the generator and discriminator) with the conditional density of VE $\tilde{p}_r$ instead of $p_r$, but we need to obtain a generator $G$ which can generate the conditional density of $p_r$ from our cGAN model. To this end, our cGAN model has the following special structure. The generator in our cGAN model, called the *vicinal* generator $\tilde{G} : \mathcal{Z} \times \mathcal{X}' \to \mathcal{Y}$, consists of two components. The first component has the auxiliary label $\boldsymbol{x}'$ as its input and produces the label $\boldsymbol{x}$ as its output with the help of $\tilde{q}_{\boldsymbol{x}'}(\boldsymbol{x})$ in (9). The second component is the *original* generator $G : \mathcal{Z} \times \mathcal{X} \to \mathcal{Y}$ having two types of inputs: one is the label $\boldsymbol{x}$ from the first component and the other is the latent variable $\boldsymbol{z}$. So the generator $\tilde{G}$ generates $\tilde{p}_g(\cdot|\boldsymbol{x}')$ to estimate the VE $\tilde{p}_r(\cdot|\boldsymbol{x}')$ of the true distribution. The main reason why we consider this special structure in $\tilde{G}$ is that it is not possible to retrieve $p_g(\cdot|\boldsymbol{x})$ from $\tilde{p}_g(\cdot|\boldsymbol{x}')$ generated by $\tilde{G}$ if it has the same structure as a legacy generator, as deriving the original distribution from its VE is ill-posed in general. Another reason is that, since the generator $G$ is implemented in $\tilde{G}$, after finishing training we can use the trained generator $G$ to generate $p_g(\cdot|\boldsymbol{x})$, which is the objective of this work. The sampling algorithm from $\tilde{G}$ is summarized in Algorithm 2.

For the discriminator of our cGAN model, it has a similar structure as the unconditional GAN discussed in (Yang & E, 2021) as follows. We assume that there exists a *parameter* function

$$a(\boldsymbol{w}, b, \boldsymbol{x}') : S_{\boldsymbol{w},b}^1 := \{(\boldsymbol{w}, b) | \boldsymbol{w} \in \mathbb{R}^{d_{\boldsymbol{y}}}, b \in \mathbb{R}, ||\boldsymbol{w}||_1 + |b|_1 \leq 1\} \times \mathcal{X}' \to \mathbb{R} \qquad (13)$$

where

$$\tilde{D}_a(\boldsymbol{y}, \boldsymbol{x}') = \mathbb{E}_{(\boldsymbol{w},b)\sim\rho_0}\left[a(\boldsymbol{w}, b, \boldsymbol{x}')\sigma(\boldsymbol{w}\cdot\boldsymbol{y}+b)\right], \tag{14}$$

$\rho_0$ is the uniform distribution in $S^1_{\boldsymbol{w},b}$, and $\sigma$ is the ReLU function. Note that **local discriminators** $\tilde{D}_{a,\boldsymbol{x}'} = \tilde{D}_a(\cdot, \boldsymbol{x}')$ form an RKHS known as **Barron Space** $\mathcal{H}$ which is equipped with the inner product

$$\langle \tilde{D}_a(\cdot, \boldsymbol{x}'), \tilde{D}_{a'}(\cdot, \boldsymbol{x}')\rangle_{\mathcal{H}} = \mathbb{E}_{(\boldsymbol{w},b)\sim\rho_0}\left[a(\boldsymbol{w}, b, \boldsymbol{x}')a'(\boldsymbol{w}, b, \boldsymbol{x}')\right] \tag{15}$$

and the reproducing kernel

$$k(\boldsymbol{y}, \boldsymbol{y}') = \mathbb{E}_{(\boldsymbol{w},b)\sim\rho_0}\left[\sigma(\boldsymbol{w}\cdot\boldsymbol{y}+b)\sigma(\boldsymbol{w}\cdot\boldsymbol{y}'+b)\right]. \tag{16}$$

For simplicity, we drop the subscript $a$ in $\tilde{D}_{a,\boldsymbol{x}'}$ from now on. Moreover, since the training of $\tilde{G}$ is equivalent to the training of $G$, we use $G$ as an argument in the loss functions below.

We now consider a cGAN loss using Wasserstein loss (Arjovsky et al., 2017) as follows:

$$V(G, \tilde{D}) = \mathbb{E}_{\boldsymbol{x}'\sim\rho(\boldsymbol{x}')}\left[\mathbb{E}_{\boldsymbol{y}\sim\tilde{p}_r(\boldsymbol{y}|\boldsymbol{x}')}[\tilde{D}(\boldsymbol{y}, \boldsymbol{x}')] - \mathbb{E}_{\boldsymbol{y}\sim\tilde{p}_g(\boldsymbol{y}|\boldsymbol{x}')}[\tilde{D}(\boldsymbol{y}, \boldsymbol{x}')]\right], \tag{17}$$

where $\rho$ is any predetermined distribution on $\mathcal{X}'$. In this paper, we further add the RKHS norm as a penalty term for regularization to the cGAN loss and consider **two-time-scale training**: We train $\tilde{D}$ to maximize the cGAN loss as well as minimizing RKHS norm of its local discriminator. It can be implemented as the maximization problem for the cGAN loss with an addtional term penalizing the average RKHS norm of a local discrminator, i.e.,

$$V'(G, \tilde{D}) = \mathbb{E}_{\boldsymbol{x}'\sim\rho(\boldsymbol{x}')}\left[\mathbb{E}_{\boldsymbol{y}\sim\tilde{p}_r(\boldsymbol{y}|\boldsymbol{x}')}[\tilde{D}(\boldsymbol{y}, \boldsymbol{x}')] - \mathbb{E}_{\boldsymbol{y}\sim\tilde{p}_g(\boldsymbol{y}|\boldsymbol{x}')}[\tilde{D}(\boldsymbol{y}, \boldsymbol{x}')] - ||\tilde{D}_{\boldsymbol{x}'}||^2_{\mathcal{H}}\right]. \tag{18}$$

and find $\tilde{D}$ that maximizes $V'(G, \tilde{D})$. After finding $\tilde{D}$, $G$ can be trained to minimize $V(G, \tilde{D})$.

For the empirical dataset, $\tilde{p}_r$ is replaced by a VE of empirical distribution $\tilde{p}_{X,Y}$ so $\tilde{D}$ is trained to maximize

$$\hat{V}'(G, \tilde{D}) = \mathbb{E}_{\boldsymbol{x}'\sim\rho(\boldsymbol{x}')}\left[\mathbb{E}_{\boldsymbol{y}\sim\tilde{p}_{X,Y}(\boldsymbol{y}|\boldsymbol{x}')}[\tilde{D}(\boldsymbol{y}, \boldsymbol{x}')] - \mathbb{E}_{\boldsymbol{y}\sim\tilde{p}_g(\boldsymbol{y}|\boldsymbol{x}')}[\tilde{D}(\boldsymbol{y}, \boldsymbol{x}')] - ||\tilde{D}_{\boldsymbol{x}'}||^2_{\mathcal{H}}\right] \tag{19}$$

and then $G$ is trained to minimize

$$\hat{V}(G, \tilde{D}) = \mathbb{E}_{\boldsymbol{x}'\sim\rho(\boldsymbol{x}')}\left[\mathbb{E}_{\boldsymbol{y}\sim\tilde{p}_{X,Y}(\boldsymbol{y}|\boldsymbol{x}')}[\tilde{D}(\boldsymbol{y}, \boldsymbol{x}')] - \mathbb{E}_{\boldsymbol{y}\sim\tilde{p}_g(\boldsymbol{y}|\boldsymbol{x}')}[\tilde{D}(\boldsymbol{y}, \boldsymbol{x}')]\right]. \tag{20}$$

## 5 ANALYSIS ON GENERALIZATION ERROR

To analyze a generalization error, we consider a cGAN with VE trained with the loss in (18) and first find the maximizer $\tilde{D}^* = \arg\max_{\tilde{D}} V'(G, \tilde{D})$. Using variational derivative, the loss is maximized when the parameter $a^*$ of the discriminator $\tilde{D}$ satisfies

$$a^*(\boldsymbol{w}, b, \boldsymbol{x}') = \frac{1}{2}\int\sigma(\boldsymbol{w}\cdot\boldsymbol{y}+b)d(\tilde{p}_r(\cdot|\boldsymbol{x}') - \tilde{p}_g(\cdot|\boldsymbol{x}'))(\boldsymbol{y}) \tag{21}$$

which results in an optimal discriminator

$$\tilde{D}^*(\boldsymbol{y}, \boldsymbol{x}') = \frac{1}{2}\int k(\boldsymbol{y}, \boldsymbol{y}')d(\tilde{p}_r(\cdot|\boldsymbol{x}') - \tilde{p}_g(\cdot|\boldsymbol{x}'))(\boldsymbol{y}') = \frac{1}{2}k*(\tilde{p}_r(\cdot|\boldsymbol{x}') - \tilde{p}_g(\cdot|\boldsymbol{x}')) \tag{22}$$

which is the convolution of the kernel and the difference between the conditional densities of the VE. Using this optimal discriminator, (17) becomes a minimization problem with respect to $G$ for the loss function

$$L(G) = \mathbb{E}_{\boldsymbol{x}'\sim\rho(\boldsymbol{x}')}\frac{1}{2}\int\int k(\boldsymbol{y}, \boldsymbol{y}')d(\tilde{p}_r(\cdot|\boldsymbol{x}') - \tilde{p}_g(\cdot|\boldsymbol{x}'))(\boldsymbol{y})d(\tilde{p}_r(\cdot|\boldsymbol{x}) - \tilde{p}_g(\cdot|\boldsymbol{x}))(\boldsymbol{y}'). \tag{23}$$

We assume that the capacity of $G$ is wide enough to handle any $\tilde{p}_g$. We then rewrite the previous equation as the optimization problem on $\tilde{p}_g$ instead:

$$L(\tilde{p}_g) = \mathbb{E}_{\boldsymbol{x}'\sim\rho(\boldsymbol{x}')}\frac{1}{2}\int\int k(\boldsymbol{y}, \boldsymbol{y}')d(\tilde{p}_r(\cdot|\boldsymbol{x}') - \tilde{p}_g(\cdot|\boldsymbol{x}'))(\boldsymbol{y})d(\tilde{p}_r(\cdot|\boldsymbol{x}) - \tilde{p}_g(\cdot|\boldsymbol{x}))(\boldsymbol{y}'). \tag{24}$$

Now assume that $\tilde{p}_g$ is trained with respect to gradient descent on itself. Then $\tilde{p}_g$ satisfies the flow equation of

$$\frac{d\tilde{p}_g^t(\boldsymbol{y}|\boldsymbol{x}')}{dt} = k * (\tilde{p}_r(\cdot|\boldsymbol{x}') - \tilde{p}_g^t(\cdot|\boldsymbol{x}')) \tag{25}$$

where $\tilde{p}_g^t(\boldsymbol{y}|\boldsymbol{x}')$ is constrained in the space of the probability density function $\Delta = \{p \in L^2(\mathcal{Y})|p \geq 0 \text{ a.e.}, \int p = 1\}$ for each $\boldsymbol{x}' \in \mathcal{X}$, using the projection $\Pi_\Delta(\tilde{p}_g^t(\boldsymbol{y}|\boldsymbol{x}')) = \tilde{p}_g^t(\boldsymbol{y}|\boldsymbol{x}')/\int \tilde{p}_g^t(\boldsymbol{y}|\boldsymbol{x}')d\boldsymbol{y}$, if necessary. Replacing $\tilde{p}_g^t(\cdot|\boldsymbol{x}') \in \Delta$ by $\mathcal{G}^t(\boldsymbol{x}') \in L^2(\mathcal{Y})$, the flow equation becomes

$$\frac{d\mathcal{G}^t(\boldsymbol{x}')}{dt} = k * (\tilde{p}_r(\cdot|\boldsymbol{x}') - \mathcal{G}^t(\boldsymbol{x}')). \tag{26}$$

We can do the same procedure using $\tilde{p}_{X,Y}$ instead of $\tilde{p}_r$, to have another trajectory $\hat{\mathcal{G}}^t(\boldsymbol{x}')$ whose flow equation is given as

$$\frac{d\hat{\mathcal{G}}^t(\boldsymbol{x}')}{dt} = k * (\tilde{p}_{X,Y}(\cdot|\boldsymbol{x}') - \hat{\mathcal{G}}^t(\boldsymbol{x}')) \tag{27}$$

We derive a generalization bound by dividing it into two terms: discrepancy between two trajectories $\hat{\mathcal{G}}^t(\boldsymbol{x}')$ and $\mathcal{G}^t(\boldsymbol{x}')$ and the gap between $\mathcal{G}^t(\boldsymbol{x}')$ and $\tilde{p}_r(\cdot|\boldsymbol{x}')$. Assuming that two trajectories share the same initial point $\mathcal{G}^0(\boldsymbol{x}') = \hat{\mathcal{G}}^0(\boldsymbol{x}')$, we can first bound the difference between $\mathcal{G}^t(\boldsymbol{x}')$ and $\hat{\mathcal{G}}^t(\boldsymbol{x}')$ by the following theorem:

**Theorem 2.** *For every $\delta > 0$ and $t > 0$, with probability $1 - \delta$ over i.i.d. sampling on $(X, Y)$,*

$$\mathbb{E}_{\boldsymbol{x}' \sim \rho(\boldsymbol{x}')} W_2(\Pi_\Delta(\hat{\mathcal{G}}^t(\boldsymbol{x}')), \Pi_\Delta(\mathcal{G}^t(\boldsymbol{x}'))) \leq \sqrt{d_{\boldsymbol{y}}}||\tilde{q}||_\infty \frac{4\sqrt{2\log 2d_{\boldsymbol{y}}} + \sqrt{2\log(2/\delta)}}{\sqrt{N}}t. \tag{28}$$

On the other hand, the trajectory $\mathcal{G}^t(\boldsymbol{x}')$ converges to the true conditional density $\tilde{p}_r(\cdot|\boldsymbol{x}')$, but we can further say that its expectation on Wasserstein-2 distance shrinks by the factor of $\sqrt{t}$:

**Theorem 3.** *For every $t > 0$,*

$$\mathbb{E}_{\boldsymbol{x}' \sim \rho(\boldsymbol{x}')} W_2(\Pi_\Delta(\mathcal{G}^t(\boldsymbol{x}')), \tilde{p}_r(\boldsymbol{y}|\boldsymbol{x}')) \leq \sqrt{d_{\boldsymbol{y}}} \frac{\mathbb{E}_{\boldsymbol{x}' \sim \rho(\boldsymbol{x}')}[||\tilde{p}_r(\boldsymbol{y}|\boldsymbol{x}') - \mathcal{G}^0(\boldsymbol{x}')||_\mathcal{H}]}{\sqrt{t}}. \tag{29}$$

Summing up two inequalities and using the fact that Wasserstein-2 distance is a metric, we have the following generalization bound on the distance between the conditional densities of the VE:

**Theorem 4** (Generalization Bound on $\tilde{p}_g$). *For every $\delta > 0$ and $t > 0$, with probability $1 - \delta$ over i.i.d. sampling on $(X, Y)$,*

$$\mathbb{E}_{\boldsymbol{x}' \sim \rho(\boldsymbol{x}')} W_2(\Pi_\Delta(\hat{\mathcal{G}}^t(\boldsymbol{x}')), \tilde{p}_r(\boldsymbol{y}|\boldsymbol{x}')) \leq$$
$$\sqrt{d_{\boldsymbol{y}}}||\tilde{q}||_\infty \frac{4\sqrt{2\log 2d_{\boldsymbol{y}}} + \sqrt{2\log(2/\delta)}}{\sqrt{N}}t + \sqrt{d_{\boldsymbol{y}}} \frac{\mathbb{E}_{\boldsymbol{x}' \sim \rho(\boldsymbol{x}')}[||\tilde{p}_r(\boldsymbol{y}|\boldsymbol{x}') - \mathcal{G}^0(\boldsymbol{x}')||_\mathcal{H}]}{\sqrt{t}}. \tag{30}$$

We can select $t^*$, an optimal value of $t$ which minimizes the right-hand side of (30), by

$$t^* = \left( \frac{\mathbb{E}_{\boldsymbol{x}' \sim \rho(\boldsymbol{x}')}[||\tilde{p}_r(\boldsymbol{y}|\boldsymbol{x}') - \mathcal{G}^0(\boldsymbol{x}')||_\mathcal{H}]}{8\sqrt{2\log 2d_{\boldsymbol{y}}} + 2\sqrt{2\log(2/\delta)}} \right)^{2/3} \frac{N^{1/3}}{||\tilde{q}||_\infty^{2/3}} \tag{31}$$

which results in the generalization error of scale $\mathcal{O}(N^{-1/6})$:

**Corollary 1** (Generalization Bound on $\tilde{p}_g$ with Early Stopping). *Let $\delta > 0$ and $t^*$ be defined as (31). Then with probability $1 - \delta$ over i.i.d. sampling on $(X, Y)$,*

$$\mathbb{E}_{\boldsymbol{x}' \sim \rho(\boldsymbol{x}')} W_2(\Pi_\Delta(\hat{\mathcal{G}}^{t^*}(\boldsymbol{x}')), \tilde{p}_r(\boldsymbol{y}|\boldsymbol{x}')) \leq \tag{32}$$
$$\frac{3}{2}\sqrt{d_{\boldsymbol{y}}} \left( 4\sqrt{2\log 2d_{\boldsymbol{y}}} + \sqrt{2\log(2/\delta)} \right)^{1/3} \left( \mathbb{E}_{\boldsymbol{x}' \sim \rho(\boldsymbol{x}')}[||\tilde{p}_r(\boldsymbol{y}|\boldsymbol{x}') - \mathcal{G}^0(\boldsymbol{x}')||_\mathcal{H}] \right)^{2/3} ||\tilde{q}||_\infty^{1/3} N^{-1/6}.$$

Therefore, we have a generated conditional densities $\tilde{p}_g(\cdot|\boldsymbol{x}') = \Pi_\Delta(\hat{\mathcal{G}}^{t^*}(\boldsymbol{x}'))$ which has a PAC-Bayesian bound of scale $\mathcal{O}(N^{-1/6})$.

As the last step, we convert the bound on the distance between conditional densities of the VE into the distance between conditional densities of the original distribution with the help of the result in the previous section. We assume that $p_r$ and $p_g$ are $L_r$- and $L_g$-label Lipschitz, respectively. Then we can construct a PAC-Bayesian bound on generalization error on $p_g$:

**Theorem 5** (Generalization Bound on $p_g$). *Let generator $G$ is trained so that the distribution $\tilde{p}_g(\boldsymbol{x}') = \Pi_\Delta(\hat{\mathcal{G}}^{t^*}(\cdot|\boldsymbol{x}'))$ is trained under the flow equation (27) and early stopping at time $t^*$ in (31). and assume that $p_r$ and $p_g$ are $L_r$ and $L_g$-label Lipschitz continuous respectively, and there exists $D > 0$ that $d_{\tilde{q}}(\boldsymbol{x}') \leq D||\tilde{q}||_\infty^{-1/d_{\boldsymbol{x}}}$. Then with probability $1 - \delta$ over i.i.d. sampling on $(X, Y)$,*

$$\mathbb{E}_{\boldsymbol{x}' \sim \rho(\boldsymbol{x}')} W_2(p_g(\cdot|m_{\tilde{q}}(\boldsymbol{x}')), p_r(\cdot|m_{\tilde{q}}(\boldsymbol{x}'))) \leq \mathcal{O}(||\tilde{q}||_\infty^{1/3} N^{-1/6}) + (L_r + L_g)D||\tilde{q}||_\infty^{-1/d_{\boldsymbol{x}}}. \quad (33)$$

Selecting the optimal scale of $||\tilde{q}||_\infty = \mathcal{O}(N^{\frac{d_{\boldsymbol{x}}}{6+2d_{\boldsymbol{x}}}})$, we finally obtain a generalization bound on $p_g$:

**Corollary 2** (Generalization Bound on $p_g$ with the optimal scale of $||\tilde{q}||_\infty$). *Under the optimal scale of $||\tilde{q}||_\infty$, we have*

$$\mathbb{E}_{\boldsymbol{x}' \sim \rho(\boldsymbol{x}')} W_2(p_g(\cdot|m_{\tilde{q}}(\boldsymbol{x}')), p_r(\cdot|m_{\tilde{q}}(\boldsymbol{x}'))) \leq \mathcal{O}(N^{-\frac{1}{6+2d_{\boldsymbol{x}}}}). \quad (34)$$

We end this section by discussing the bounds and the assumption in our analysis:

- Note that we do not consider any assumption on $\rho$ except for being a distribution on $\mathcal{X}'$, so the bound in Corollary 2 can be applied over various distributions on $\mathcal{X}$ as well. For example, when $\tilde{q}$ is a perturbation as in the examples, we can simply set $\rho(\boldsymbol{x}') = p_r(\boldsymbol{x}')$ to get a generalization error of $\mathbb{E}_{\boldsymbol{x} \sim p_r(\boldsymbol{x})} W_2(p_g(\cdot|\boldsymbol{x}), p_r(\cdot|\boldsymbol{x}))$.

- The bound of $\mathcal{O}(N^{-\frac{1}{6+2d_{\boldsymbol{x}}}})$ is independent of the output dimensionality $d_{\boldsymbol{y}}$, but it is not independent of the label dimensionality $d_{\boldsymbol{x}}$ and hence still vulnerable for the Curse of Dimensionality in high $d_{\boldsymbol{x}}$. However, $d_{\boldsymbol{x}}$ is usually assumed to be a very small integer or even 1 for most of its applications.

- Assuming label Lipschitzness on $p_r$ and $p_g$ is crucial to bound the distance between the conditional densities of its original distribution. Even though the Lipschitzness on $p_r$ cannot be verified in practice, we can easily enforce label Lipschitzness on $p_g$ by giving Lipschitz continuity on the generator $G$ with the methods like spectral normalization (SN) (Miyato et al., 2018) or Lipschitz constant constraint (LCC) Gouk et al. (2020) on its layers.

## 6 EXPERIMENTS

To validate our analysis in the previous section, we construct a simple toy example $p_r$ with

$$p_r(\boldsymbol{x}) \sim \mathcal{N}(0, 1), p_r(\boldsymbol{y}|\boldsymbol{x}) \sim (s(\boldsymbol{x}), \cdots, s(\boldsymbol{x}))/2 + \mathcal{U}([0, 1/2]^{d_{\boldsymbol{y}}}), \quad (35)$$

where $s(\boldsymbol{x}) = 1/(1 + e^{-\boldsymbol{x}})$ is a sigmoid function and $\mathcal{U}$ means a uniform distribution. Note that the conditional distribution $p_r(\boldsymbol{y}|x)$ is $\sqrt{d_{\boldsymbol{y}}}/8$-label Lipschitz continuous and $\mathcal{Y} \subset [0, 1]^{d_{\boldsymbol{y}}}$. Now we try to predict the conditional density $p_r(\cdot|x)$ from the empirical distribution $p_{X,Y}(\cdot|x)$ of $N$ samples with the following two cGAN models:

- **cGAN model (Baseline cGAN):** It uses the minimax training with the standard Wasserstein GAN loss (5) for 3000 epochs and the layers of $D$ are normalized by using spectral normalization.

- **VE-cGAN model (Proposed):** It uses a VE defined by the perturbation on $\boldsymbol{x}$ whose inverse auxiliary distribution is $\tilde{q}_{\boldsymbol{x}'}(\boldsymbol{x}) = \mathcal{N}(\boldsymbol{x}', \sigma^2)$, the penalized loss in (19) with $\rho(\boldsymbol{x}') = \mathcal{N}(0, 1)$, early stopping at epoch $\lfloor 100N^{1/4} \rfloor$, and the optimal value $\sigma = 0.4N^{-1/8}$.

Two cGAN models have the same structure including generator, discriminator, latent space, initialization, and hyperparameters. Note that the same discriminator can be used for both models because

Table 1: Optimal Values of $\alpha$

| $d_{\boldsymbol{y}}$ | 4 | 8 | 16 | 32 | 64 | 128 |
|---|---|---|---|---|---|---|
| Baseline cGAN | 0.393 | 0.223 | 0.107 | 0.076 | 0.030 | 0.009 |
| VE-cGAN | 0.501 | 0.423 | 0.348 | 0.289 | 0.252 | 0.235 |

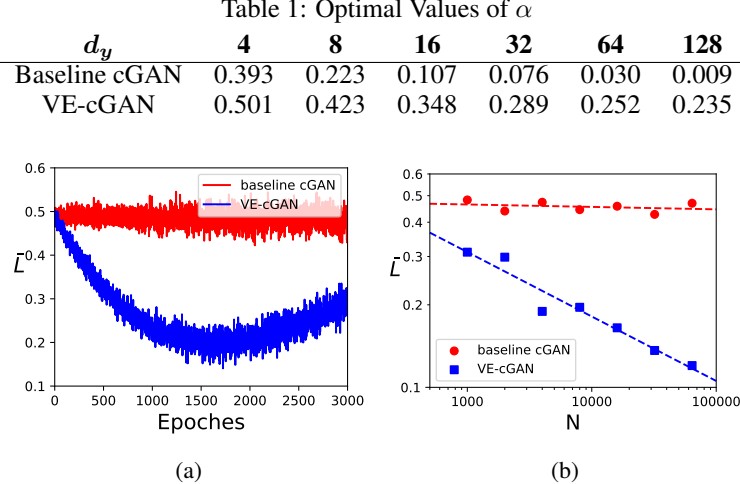

(a)                                    (b)

Figure 1: (a) Training curve of $\bar{L}$ (averaged over 20 random seeds) for each epoch when $d_{\boldsymbol{y}} = 128$ and $N = 64000$. The VE-cGAN model is trained additionally until 3000 epoches for comparison with the baseline cGAN. (b) Graph between $N$ and $\bar{L}$ in the case of $d_{\boldsymbol{y}} = 128$. The dashed lines represent the graphs of $\log \bar{L} = \log \bar{L}_0 - \alpha \log N$ for two cGAN models.

$\mathcal{X} = \mathcal{X}'$ in this example. For each pair of $N \in \{1000, 2000, 4000, 8000, 16000, 32000, 64000\}$ and $d_{\boldsymbol{y}} \in \{4, 8, 16, 32, 64, 128\}$, we train two cGAN models with 20 different samplings on $(X, Y)$. Instead of the true generalization error $\mathbb{E}_{x \sim \mathcal{N}(0,1)} W_2(p_r(\boldsymbol{y}|x), p_g(\boldsymbol{y}|x))$ which is not tractable in high dimension, we evaluate $L(p_g) = \mathbb{E}_{x \sim \mathcal{N}(0,1)} W_2(\bar{p}_r(\boldsymbol{y}|x), \bar{p}_g(\boldsymbol{y}|x))$ where $\bar{p}(\boldsymbol{y}|x)$ denotes the distribution of the norm $\|\boldsymbol{y} - (s(\boldsymbol{x}), \cdots, s(\boldsymbol{x}))/2\|$ for $\boldsymbol{y} \sim p(\boldsymbol{y}|x)$. After averaging $L(p_g)$ over 20 different samplings (say the average $\bar{L}$) for each pair of $N$ and $d_{\boldsymbol{y}}$, we compare two cGAN models by interpolating them with the curve $\bar{L} = \bar{L}_0 N^{-\alpha}$ for each value of $d_{\boldsymbol{y}}$, which is done by a linear regression with $\log \bar{L} = \log \bar{L}_0 - \alpha \log N$. The optimal values of $\alpha$ from the linear regression are provided in Table 1.

From Table 1 we see that the average $\bar{L}$ of the VE-cGAN model decays with respect to $N$ much faster than that of the baseline cGAN model. Figure 1 shows how much $\bar{L}$ changes during the training for each cGAN model and how it decays as a function of $N$ in the case of $d_{\boldsymbol{y}} = 128$. In Figure 1(a), the VE-cGAN model shows a rapid decrease in $\bar{L}$ up to a certain epoch followed by an increase, compared with the baseline cGAN which barely learns anything. The VE-cGAN model achieves its optimum around epoch 1600, which approximately agrees with our theoretical result $\lfloor 100 N^{1/4} \rfloor = 1590$ on early stopping. Figure 1(b) shows that $\bar{L}$ decays with respect to $N$ in the VE-cGAN model whereas it does not show any significant difference in the baseline cGAN. From Table 1 and Figure 1, we can conclude that the VE-cGAN model has a good generalization performance even in high $d_{\boldsymbol{y}}$, compared with the baseline cGAN model. Similar trends are observed for other values of $d_{\boldsymbol{y}}$ as well. More experimental details and results are provided in Appendix B.

## 7 CONCLUSIONS AND FUTURE WORKS

We introduced the concept of the Vicinal Estimate (VE) of a distribution and proposed a new cGAN model with VE for estimating conditional density. For the proposed cGAN model we derived a generalization error bound that is independent of the output dimensionality, which means that the proposed cGAN model can escape from both the *Curse of Dimensionality* and *Lack of Conditional Samples* issues. We conducted an experiment to demonstrate that the generalization error of the proposed cGAN decays effectively in high dimension. We believe that our work contributes to understanding how the VRM-based method can improve the generalization performance of cGANs.

One of the main limitations of our analysis is that the Barron space of discriminators is not large enough to handle more complex cGAN models. However, we believe that, if we can derive generalization errors for some complex unconditional GANs, then we can also derive generalization errors for the corresponding cGANs with the same approach for unconditional GANs as we did in this work.

## 8 REPRODUCIBILITY STATEMENT

All the codes used in the experiments in section 6 will be uploaded online. In addition, we provide in Appendix B the detailed information on the parameters and the model structure to make it easy for anyone to reproduce the results.

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

# A  PROOF OF THE THEOREMS

## A.1  PROOF OF LEMMA 1

Assume that $p$ is $L$-label Lipschtiz continuous. Then

$$
\begin{aligned}
&W_2(\tilde{p}(\cdot|\boldsymbol{x}'), p(\cdot|m_{\tilde{q}}(\boldsymbol{x}'))) \\
&= W_2\left(\int_{\mathcal{X}} p(\cdot|\boldsymbol{x})\tilde{q}_{\boldsymbol{x}'}(\boldsymbol{x})d\boldsymbol{x}, p(\cdot|m_{\tilde{q}}(\boldsymbol{x}'))\right) \\
&= W_2\left(\int_{\mathcal{X}} p(\cdot|\boldsymbol{x})\tilde{q}_{\boldsymbol{x}'}(\boldsymbol{x})d\boldsymbol{x}, \int_{\mathcal{X}} p(\cdot|m_{\tilde{q}}(\boldsymbol{x}'))\tilde{q}_{\boldsymbol{x}'}(\boldsymbol{x})d\boldsymbol{x}\right) \\
&\leq \sqrt{\int_{\mathcal{X}} (W_2(p(\cdot|\boldsymbol{x}), p(\cdot|m_{\tilde{q}}(\boldsymbol{x}'))))^2 \tilde{q}_{\boldsymbol{x}'}(\boldsymbol{x})d\boldsymbol{x}} && (*) \\
&\leq L\sqrt{\int_{\mathcal{X}} ||\boldsymbol{x} - m_{\tilde{q}}(\boldsymbol{x}'))||^2 \tilde{q}_{\boldsymbol{x}'}(\boldsymbol{x})d\boldsymbol{x}} && (**) \\
&= L\sqrt{\mathbb{E}_{\tilde{q}_{\boldsymbol{x}'}(\boldsymbol{x})}[||\boldsymbol{x} - m_{\tilde{q}}(\boldsymbol{x}'))||^2]}. && (36)
\end{aligned}
$$

Here, (**) holds by simply applying the $L$-label Lipschitz continuity of $p$. To show (*) let $\epsilon > 0$. From the definition of $W_2(p(\cdot|\boldsymbol{x}), p(\cdot|m_{\tilde{q}}(\boldsymbol{x}')))$, for each $\boldsymbol{x}$ we have a joint probability $\gamma_{\boldsymbol{x}}(\boldsymbol{y}_1, \boldsymbol{y}_2)$ of $p(\cdot|\boldsymbol{x})$ and $p(\cdot|m_{\tilde{q}}(\boldsymbol{x}'))$ such that

$$
\mathbb{E}_{(\boldsymbol{y}_1, \boldsymbol{y}_2) \sim \gamma_{\boldsymbol{x}}} ||\boldsymbol{y}_1 - \boldsymbol{y}_2||^2 < W_2(p(\cdot|\boldsymbol{x}), p(\cdot|m_{\tilde{q}}(\boldsymbol{x}')))^2 + \epsilon. \tag{37}
$$

We next construct a joint probability $\gamma(\boldsymbol{y}_1, \boldsymbol{y}_2)$ between $\tilde{p}(\cdot|\boldsymbol{x}')$ and $p(\cdot|m_{\tilde{q}}(\boldsymbol{x}'))$ by $\gamma(\boldsymbol{y}_1, \boldsymbol{y}_2) = \int_{\mathcal{X}} \gamma_{\boldsymbol{x}}(\boldsymbol{y}_1, \boldsymbol{y}_2)\tilde{q}_{\boldsymbol{x}'}(\boldsymbol{x})d\boldsymbol{x}$. We then have

$$
\begin{aligned}
(W_2(\tilde{p}(\cdot|\boldsymbol{x}), p(\cdot|m_{\tilde{q}}(\boldsymbol{x}'))))^2 &\leq \mathbb{E}_{(\boldsymbol{y}_1, \boldsymbol{y}_2) \sim \gamma} ||\boldsymbol{y}_1 - \boldsymbol{y}_2||^2 \\
&= \int_{\mathcal{X}} \mathbb{E}_{(\boldsymbol{y}_1, \boldsymbol{y}_2) \sim \gamma_{\boldsymbol{x}}} ||\boldsymbol{y}_1 - \boldsymbol{y}_2||^2 \tilde{q}_{\boldsymbol{x}'}(\boldsymbol{x})d\boldsymbol{x} \\
&\leq \int_{\mathcal{X}} (W_2(p(\cdot|\boldsymbol{x}), p(\cdot|m_{\tilde{q}}(\boldsymbol{x}'))))^2 \tilde{q}_{\boldsymbol{x}'}(\boldsymbol{x})d\boldsymbol{x} + \epsilon \tag{38}
\end{aligned}
$$

so we can take the limit of $\epsilon \to 0$ and take the square root to obtain (*).

## A.2  PROOF OF THEOREM 1

Let $\bar{q}(\boldsymbol{x}^*)$ be the distribution of the random variable $\boldsymbol{x}^* = (\boldsymbol{x} - m_{\tilde{q}^L}(\boldsymbol{x}'))/L$. Then $\tilde{q}_{\boldsymbol{x}'}^L(\boldsymbol{x})$ can be written as

$$
\tilde{q}_{\boldsymbol{x}'}^L(\boldsymbol{x}) = L^{-d_{\boldsymbol{x}}} \bar{q}\left(\frac{\boldsymbol{x} - m_{\tilde{q}^L}(\boldsymbol{x}')}{L}\right). \tag{39}
$$

We then have $||\tilde{q}^L||_{\infty} = L^{-d_{\boldsymbol{x}}}||\bar{q}||_{\infty}$ and $d_{\tilde{q}^L}(\boldsymbol{x}') = Ld_{\bar{q}}$ where $d_{\bar{q}} = \sqrt{\int_{\mathcal{X}} ||\boldsymbol{x}^*||^2 \bar{q}(\boldsymbol{x}^*)d\boldsymbol{x}^*}$ is the $\ell^2$-average distance of $\boldsymbol{x}^*$. Therefore, we have

$$
\begin{aligned}
d_{\tilde{q}^L}(\boldsymbol{x}') &= d_{\bar{q}}||\bar{q}||_{\infty}^{1/d_{\boldsymbol{x}}}||\tilde{q}^L||_{\infty}^{-1/d_{\boldsymbol{x}}} \tag{40} \\
&= C||\tilde{q}^L||_{\infty}^{-1/d_{\boldsymbol{x}}} \tag{41}
\end{aligned}
$$

for all $L > 0$ where $C = d_{\bar{q}}||\tilde{q}||_{\infty}^{1/d_{\boldsymbol{x}}}$.

## A.3  PROOF OF THEOREM 2

We first use the lemma from (Yang & E, 2021):

**Lemma 2.** *For any $p, q \in L^2(\mathcal{Y})$, $W_2(\Pi_{\Delta}(p), \Pi_{\Delta}(q)) = \sqrt{d_{\boldsymbol{y}}}||p - q||_{L^2(\mathcal{Y})}$.*

From Lemma 2, it is enough to show the inequality on $\mathbb{E}_{\boldsymbol{x}'\sim\rho(\boldsymbol{x}')}||\hat{\mathcal{G}}^t(\boldsymbol{x}') - \mathcal{G}^t(\boldsymbol{x}')||_{L^2(\mathcal{Y})}$. Consider its derivative with respect to $t$:

$$\frac{d}{dt}\mathbb{E}_{\boldsymbol{x}'\sim\rho(\boldsymbol{x}')}||\hat{\mathcal{G}}^t(\boldsymbol{x}') - \mathcal{G}^t(\boldsymbol{x}')||_{L^2(\mathcal{Y})}$$

$$= \mathbb{E}_{\boldsymbol{x}'\sim\rho(\boldsymbol{x}')}\left\langle \frac{\hat{\mathcal{G}}^t(\boldsymbol{x}') - \mathcal{G}^t(\boldsymbol{x}')}{||\hat{\mathcal{G}}^t(\boldsymbol{x}') - \mathcal{G}^t(\boldsymbol{x}')||}, \frac{d\hat{\mathcal{G}}^t(\boldsymbol{x}')}{dt} - \frac{d\mathcal{G}^t(\boldsymbol{x}')}{dt} \right\rangle_{L^2(\mathcal{Y})}$$

$$= \mathbb{E}_{\boldsymbol{x}'\sim\rho(\boldsymbol{x}')}\left\langle \frac{\hat{\mathcal{G}}^t(\boldsymbol{x}') - \mathcal{G}^t(\boldsymbol{x}')}{||\hat{\mathcal{G}}^t(\boldsymbol{x}') - \mathcal{G}^t(\boldsymbol{x}')||}, k*(\tilde{p}_{X,Y}(\cdot|\boldsymbol{x}') - \hat{\mathcal{G}}^t(\boldsymbol{x}')) - k*(\tilde{p}_r(\cdot|\boldsymbol{x}') - \mathcal{G}^t(\boldsymbol{x}')) \right\rangle_{L^2(\mathcal{Y})}$$

$$= \mathbb{E}_{\boldsymbol{x}'\sim\rho(\boldsymbol{x}')}\left\langle \frac{\hat{\mathcal{G}}^t(\boldsymbol{x}') - \mathcal{G}^t(\boldsymbol{x}')}{||\hat{\mathcal{G}}^t(\boldsymbol{x}') - \mathcal{G}^t(\boldsymbol{x}')||}, k*(\tilde{p}_{X,Y}(\cdot|\boldsymbol{x}') - \tilde{p}_r(\cdot|\boldsymbol{x}')) - k*(\hat{\mathcal{G}}^t(\boldsymbol{x}') - \mathcal{G}^t(\boldsymbol{x}')) \right\rangle_{L^2(\mathcal{Y})}$$

$$\leq \mathbb{E}_{\boldsymbol{x}'\sim\rho(\boldsymbol{x}')}\left\langle \frac{\hat{\mathcal{G}}^t(\boldsymbol{x}') - \mathcal{G}^t(\boldsymbol{x}')}{||\hat{\mathcal{G}}^t(\boldsymbol{x}') - \mathcal{G}^t(\boldsymbol{x}')||}, k*(\tilde{p}_{X,Y}(\cdot|\boldsymbol{x}') - \tilde{p}_r(\cdot|\boldsymbol{x}')) \right\rangle_{L^2(\mathcal{Y})}$$

$$\leq \mathbb{E}_{\boldsymbol{x}'\sim\rho(\boldsymbol{x}')}||k*(\tilde{p}_{X,Y}(\cdot|\boldsymbol{x}') - \tilde{p}_r(\cdot|\boldsymbol{x}'))||_{L^2(\mathcal{Y})}$$

$$\leq \mathbb{E}_{\boldsymbol{x}'\sim\rho(\boldsymbol{x}')}\sup_{\boldsymbol{y}\in\mathcal{Y}}\left|\mathbb{E}_{(\boldsymbol{w},b)\sim\rho_0}\left[\sigma(\boldsymbol{w}\cdot\boldsymbol{y}+b)\int\sigma(\boldsymbol{w}\cdot\boldsymbol{y}'+b)d(\tilde{p}_{X,Y}(\cdot|\boldsymbol{x}') - \tilde{p}_r(\cdot|\boldsymbol{x}'))(\boldsymbol{y}')\right]\right|$$

$$\leq \mathbb{E}_{\boldsymbol{x}'\sim\rho(\boldsymbol{x}')}\sup_{S^1_{\boldsymbol{w},b}}\left|\int\sigma(\boldsymbol{w}\cdot\boldsymbol{y}+b)d(\tilde{p}_{X,Y}(\cdot|\boldsymbol{x}') - \tilde{p}_r(\cdot|\boldsymbol{x}'))(\boldsymbol{y})\right| \tag{*}$$

Define a function $L(\boldsymbol{x},\boldsymbol{y},\boldsymbol{x}') = \sigma(\boldsymbol{w}\cdot\boldsymbol{y}+b)\tilde{q}_{\boldsymbol{x}'}(\boldsymbol{x})$. Then we have

$$\int\sigma(\boldsymbol{w}\cdot\boldsymbol{y}+b)d(\tilde{p}_{X,Y}(\cdot|\boldsymbol{x}'))(\boldsymbol{y}) = \frac{1}{N}\sum_{i=1}^{N}L(\boldsymbol{x}_i,\boldsymbol{y}_i,\boldsymbol{x}') \tag{42}$$

and

$$\int\sigma(\boldsymbol{w}\cdot\boldsymbol{y}+b)d(\tilde{p}_r(\cdot|\boldsymbol{x}'))(\boldsymbol{y}) = \frac{1}{N}\mathbb{E}_{(\boldsymbol{x},\boldsymbol{y})\sim p_r}L(\boldsymbol{x},\boldsymbol{y},\boldsymbol{x}') \tag{43}$$

which makes

$$(*) = \mathbb{E}_{\boldsymbol{x}'\sim\rho(\boldsymbol{x}')}\sup_{S^1_{\boldsymbol{w},b}}\left|\frac{1}{N}\sum_{i=1}^{N}L(\boldsymbol{x}_i,\boldsymbol{y}_i,\boldsymbol{x}') - \mathbb{E}_{(\boldsymbol{x},\boldsymbol{y})\sim p_r}L(\boldsymbol{x},\boldsymbol{y},\boldsymbol{x}')\right| \tag{44}$$

and it is the actually the average of the maximal difference between the empirical average and real average of the function $L(\boldsymbol{x}_i,\boldsymbol{y}_i,\boldsymbol{x}')$, whereas we clearly have $|L| \leq ||\tilde{q}||_\infty$. Now, we use the inequality on Ramedacher complexity like Theorem 6 in (E et al., 2021) to obtain

$$(*) \leq ||\tilde{q}||_\infty\left(2\mathrm{Rad}_N(||D||_{\mathcal{H}}\leq 1) + \frac{\sqrt{2\log(2/\delta)}}{\sqrt{N}}\right) \leq ||\tilde{q}||_\infty\frac{4\sqrt{2\log 2d_{\boldsymbol{y}}} + \sqrt{2\log(2/\delta)}}{\sqrt{N}} \tag{45}$$

with probability $1-\delta$ over i.i.d. sampling on $(X,Y)$. Clearly we have $\hat{\mathcal{G}}^0 = \mathcal{G}^0$ from having the same initial point, so we can integrate it from 0 to $t$ and apply Lemma 2 to obtain Theorem 2 (Note that the bound on $(*)$ is independent f $t$).

## A.4 PROOF OF THEOREM 3

Fix $\boldsymbol{x}' \in \mathcal{X}'$. Then for each $a \in L^2(\rho_0)$, define a function

$$f_a(\boldsymbol{y},\boldsymbol{x}') = \mathbb{E}_{(\boldsymbol{w},b)\sim\rho_0}[a(\boldsymbol{w},b,\boldsymbol{x}')\sigma(\boldsymbol{w}\cdot\boldsymbol{y}+b)] \in L^2(\mathcal{Y}). \tag{46}$$

If $||\tilde{p}_r(\boldsymbol{y}|\boldsymbol{x}') - \mathcal{G}^0(\boldsymbol{x}')||_{\mathcal{H}} < \infty$, we can choose $a_0$ such that $f_{a_0}(\boldsymbol{y},\boldsymbol{x}') = \tilde{p}_r(\boldsymbol{y}|\boldsymbol{x}') - \mathcal{G}^0(\boldsymbol{x}')$ for every $\boldsymbol{y}$. Now fix $\boldsymbol{x}'$ and consider training $a_t(\cdot,\boldsymbol{x}')$ by gradient flow with initialization $a_0(\cdot,\boldsymbol{x}')$ on the loss

$$\min_a \Gamma(a) = \frac{1}{2}||f_a(\boldsymbol{y},\boldsymbol{x}')||^2_{L^2(\mathcal{Y})}. \tag{47}$$

which implies

$$\frac{d}{dt}a_t(\boldsymbol{w}, b, \boldsymbol{x}') = -\frac{\delta\Gamma(a_t)}{\delta a_t(\boldsymbol{w}, b, \boldsymbol{x}')} = -\int f_{a_t}(\boldsymbol{y}', \boldsymbol{x}')\sigma(\boldsymbol{w}\cdot\boldsymbol{y}' + b)d\boldsymbol{y}' \tag{48}$$

The function $f_{a_t}$ evolves by

$$\frac{d}{dt}f_{a_t}(\boldsymbol{y}, \boldsymbol{x}') = \mathbb{E}_{(\boldsymbol{w},b)\sim\rho_0}\left[\frac{d}{dt}a_t(\boldsymbol{w}, b, \boldsymbol{x}')\sigma(\boldsymbol{w}\cdot\boldsymbol{y} + b)\right]$$

$$= \mathbb{E}_{(\boldsymbol{w},b)\sim\rho_0}\left[-\int f_{a_t}(\boldsymbol{y}', \boldsymbol{x}')\sigma(\boldsymbol{w}\cdot\boldsymbol{y}' + b)d\boldsymbol{y}'\sigma(\boldsymbol{w}\cdot\boldsymbol{y} + b)\right]$$

$$= -\int f_{a_t}(\boldsymbol{y}', \boldsymbol{x}')k(\boldsymbol{y}, \boldsymbol{y}')d\boldsymbol{y}' = -k * f_{a_t}(\boldsymbol{y}, \boldsymbol{x}')$$

Therefore, the dynamics of $f_{a_t}$ has the same rule as the function $\tilde{p}_r(\boldsymbol{y}|\boldsymbol{x}') - \mathcal{G}^t(\boldsymbol{x}')$ where two functions are the same at $t = 0$. Therefore, we have $f_{a_t} = \tilde{p}_r(\boldsymbol{y}|\boldsymbol{x}') - \mathcal{G}^t(\boldsymbol{x}')$ for every $t > 0$. Note that $\Gamma(a)$ is convex with respect to $a$, therefore we have

$$\frac{d}{dt}||a_t||^2_{\rho_0} = \left\langle 2a_t, \frac{da_t}{dt}\right\rangle_{\rho_0} = 2\left\langle a_t, -\frac{\delta\Gamma(a_t)}{\delta a_t}\right\rangle_{\rho_0} \leq -2\Gamma(a_t) = -||f_{a_t}||^2_{L^2(\mathcal{Y})} \tag{49}$$

and hence

$$||f_{a_t}||^2_{L^2(\mathcal{Y})} \leq -\frac{d}{dt}||a_t||^2_{\rho_0}. \tag{50}$$

Integrating both sides over $[0, t]$ yields

$$\int_0^t ||f_{a_u}||^2_{L^2(\mathcal{Y})}du \leq ||a_0||^2_{\rho_0} - ||a_t||^2_{\rho_0}. \tag{51}$$

Therefore, we have

$$t||f_{a_t}||^2_{L^2(\mathcal{Y})} \leq \int_0^t ||f_{a_u}||^2_{L^2(\mathcal{Y})}du \leq ||a_0||^2_{\rho_0} - ||a_t||^2_{\rho_0} \leq ||a_0||^2_{\rho_0} \tag{52}$$

as $||f_{a_t}||^2_{L^2(\mathcal{Y})} = \frac{1}{2}\Gamma(a_t)$ monotonically decreases with respect to $t$. It then follows that

$$||\mathcal{G}^t(\boldsymbol{x}') - \tilde{p}_r(\boldsymbol{y}|\boldsymbol{x}')||^2_{L^2(\mathcal{Y})} = ||f_{a_t}||^2_{L^2(\mathcal{Y})} \leq \frac{||a_0||^2_{\rho_0}}{t} = \frac{||\tilde{p}_r(\boldsymbol{y}|\boldsymbol{x}') - \mathcal{G}^0(\boldsymbol{x}')||^2_{\mathcal{H}}}{t} \tag{53}$$

and hence

$$||\mathcal{G}^t(\boldsymbol{x}') - \tilde{p}_r(\boldsymbol{y}|\boldsymbol{x}')||_{L^2(\mathcal{Y})} \leq \frac{||\tilde{p}_r(\boldsymbol{y}|\boldsymbol{x}') - \mathcal{G}^0(\boldsymbol{x}')||_{\mathcal{H}}}{\sqrt{t}}. \tag{54}$$

Now applying Lemma 2 and integrating over $\rho(\boldsymbol{x}')$, we obtain Theorem 3.

## A.5 Proof of Theorem 5

From Lemma 1 and $W_2$ being a metric, we have

$$\mathbb{E}_{\boldsymbol{x}'\sim\rho(\boldsymbol{x}')}W_2(p_g(\cdot|m_{\tilde{q}}(\boldsymbol{x}')), p_r(\cdot|m_{\tilde{q}}(\boldsymbol{x}')))$$

$$\leq \mathbb{E}_{\boldsymbol{x}'\sim\rho(\boldsymbol{x}')}\left[W_2(\tilde{p}_g(\cdot|\boldsymbol{x}'), \tilde{p}_r(\cdot|\boldsymbol{x}')) + W_2(\tilde{p}_g(\cdot|\boldsymbol{x}'), p_g(\cdot|m_{\tilde{q}}(\boldsymbol{x}'))) + W_2(\tilde{p}_r(\cdot|\boldsymbol{x}'), p_r(\cdot|m_{\tilde{q}}(\boldsymbol{x}')))\right]$$

$$\leq \mathbb{E}_{\boldsymbol{x}'\sim\rho(\boldsymbol{x}')}\left[W_2(\tilde{p}_g(\cdot|\boldsymbol{x}'), \tilde{p}_r(\cdot|\boldsymbol{x}')) + (L_g + L_r)d_{\tilde{q}}(\boldsymbol{x}')\right]$$

$$\leq \mathbb{E}_{\boldsymbol{x}'\sim\rho(\boldsymbol{x}')}\left[W_2(\tilde{p}_g(\cdot|\boldsymbol{x}'), \tilde{p}_r(\cdot|\boldsymbol{x}'))\right] + (L_g + L_r)D_{\tilde{q}}||\tilde{q}||^{-1/d_{\boldsymbol{x}}}_\infty \tag{*}$$

Then (33) can be obtained by applying Corollary 1 to (*).

Table 2: Generator structure of two cGAN models. fc($n$) denotes a fully-connected layer having the output of dimension $n$. (SN) after a layer means that spectral normalization is applied to the layer.

| Input: $\mathbf{z} \in \mathbb{R}^{d_{\boldsymbol{y}}}, \boldsymbol{x} \in \mathbb{R}$ |
| --- |
| fc(128) (SN), ReLU |
| fc(128) (SN), ReLU |
| fc(128) (SN), ReLU |
| fc(128) (SN), ReLU |
| fc(128) (SN), ReLU |
| fc($d_{\boldsymbol{y}}$) (SN), Logistic |
| Output: $\mathbf{y} \in \mathbb{R}^{d_{\boldsymbol{y}}}$ |

Table 3: Structure for the parameter function $a$. fc($n$) denotes a fully-connected layer having the output of dimension $n$. (SN) after a layer means that spectral normalization is applied to the layer.

| Input: $\mathbf{w} \in \mathbb{R}^{d_{\boldsymbol{y}}}, b \in \mathbb{R}, \boldsymbol{x}'(\text{or } \boldsymbol{x}) \in \mathbb{R}$ |
| --- |
| fc(128) (SN), ReLU |
| fc(128) (SN), ReLU |
| fc(128) (SN), ReLU |
| fc(128) (SN), ReLU |
| fc(1) (SN) |
| Output: $a(\boldsymbol{w}, b, \boldsymbol{x}'(\text{or } \boldsymbol{x})) \in \mathbb{R}^{d_{\boldsymbol{y}}}$ |

## B  EXPERIMENTAL DETAILS AND ANALYSIS

In this section we provide the details and results of our experiments in section 6. We assume that the latent variable $\boldsymbol{z}$ follows a unit normal distribution in $\mathbb{R}^{d_{\boldsymbol{y}}}$. In Table 2 we provide the network architecture that we adopt in the experiment for both cGAN models. Note that spectral normalization is applied to each layer so that the generated distributions are label Lipschitz.

The Barron Space, described in (14), is used as the discriminator of the VE-cGAN model. Unlike the previous work in (Yang & E, 2021) which approximates the Barron space as a two-layer feed-forward network, we introduce a new implementation which is directly derived from (14). Instead of the discriminator itself, we model the parameter function $a(\boldsymbol{w}, b, \boldsymbol{x}')$ as a neural network whose structure is provided in Table 3. Once we have $a$, $\tilde{D}(\boldsymbol{y}, \boldsymbol{x}')$ can be estimated by (14) by using a sampling $(\boldsymbol{w}, b)$ of size 2048 from $S^1_{\boldsymbol{w}, b}$. The discriminator of the baseline cGAN also uses the same Barron space structure where the auxiliary label $\boldsymbol{x}'$ is replaced by the label $\boldsymbol{x}$ (we have $\mathcal{X} = \mathcal{X}'$ from $\tilde{q}$ being a perturbation), which results in the formula of

$$D_a(\boldsymbol{y}, \boldsymbol{x}) = \mathbb{E}_{(\boldsymbol{w}, b) \sim \rho_0}[a(\boldsymbol{w}, b, \boldsymbol{x}) \sigma(\boldsymbol{w} \cdot \boldsymbol{y} + b)]. \tag{55}$$

Note that the resulting discriminator in both models is 1-Lipschitz, meaning that the baseline cGAN becomes a Wasserstein cGAN. Both cGAN models are trained with a constant learning rate ($10^{-3}$ for the discriminator and $10^{-4}$ for the generator to demonstrate two-time-scale learning), a batch size of 512, and the Adam optimizer. However, the total numbers of training epochs are different (3000 in the baseline cGAN model, $\lfloor 100 N^{1/4} \rfloor$ for the VE-cGAN model) for both models.

We lastly provide additional results to support Table 1 and to verify our claim in section 6 that similar trends are observed for other values of $d_{\boldsymbol{y}}$ as well. Figure 2 depicts the graphs between $N$ and $\bar{L}$ for various values of $d_{\boldsymbol{y}}$. From the figure we observe that $\bar{L}$ decays effectively in the VE-cGAN model for all values of $d_{\boldsymbol{y}}$ including high $d_{\boldsymbol{y}}$ whereas it fails to decay effectively for high $d_{\boldsymbol{y}}$ in the baseline cGAN model.

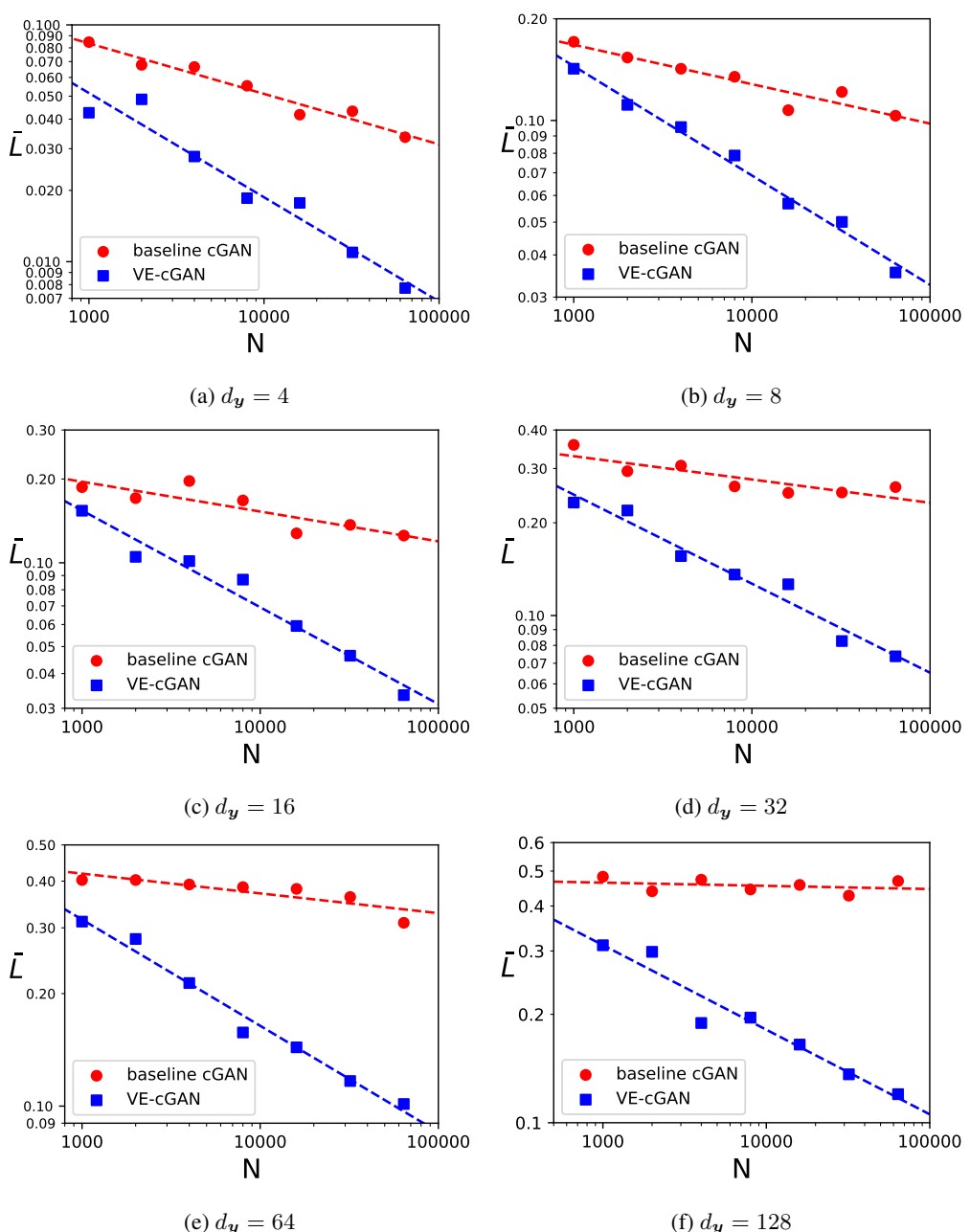

Figure 2: Graphs between $N$ and $\bar{L}$ for various values of $d_{\boldsymbol{y}}$. The dashed lines represent the graphs of $\log \bar{L} = \log \bar{L}_0 - \alpha \log N$ for the cGAN models.

