# OpenReview forum: "Improved Generalization of cGAN using Vicinal Estimation and Early Stopping"
_ICLR.cc/2024/Conference — Submitted to ICLR 2024_

### Official Review · Reviewer_JCHw · 2023-10-29

**Soundness:** 2 fair
**Presentation:** 2 fair
**Contribution:** 2 fair
**Rating:** 3
**Confidence:** 2

**Summary:**

The paper improves conditional GAN learning using Vicinal Estimation.

**Strengths:**

The idea to introduce a auxiliary distribution is interesting.

**Weaknesses:**

I was still confused about how the auxiliary distribution helps after reading the paper, because I'm not familiar with related works.

**Questions:**

The idea to use an auxiliary distribution is similar to importance sampling, are they related somehow?

---

> ### Author Response · Authors · 2023-11-16
> **Response to Reviewer JCHw**
>
> We appreciate the reviewer **JCHw** for the detailed comments and hope to address all of the questions and concerns below.
>
> > I was still confused about how the auxiliary distribution helps after reading the paper, because I'm not familiar with related works.
>
> There are two main difficulties widely known in the literature in obtaining the generalization error of cGANs: Curse of Dimensionality and Lack of Conditional Sample. In one of the related works, (Weinen & E, 2021), a framework to solve Curse of Dimensionality in unconditional GAN has been published, but it is not possible to apply it to conditional GAN as it is. Technically, a conditional GAN is a collection of unconditional GANs for all label values, but the number of samples assigned to each of these unconditional GANs is extremely small compared to the total number of samples.
>
> The intuitive explanation for the contribution of vicinal estimation (auxiliary distribution) to this problem is that, unlike the original empirical distribution, its VE provides sufficiently many conditional samples for each (auxiliary) label value. A more mathematical explanation is that the case $p\_r(\cdot|\mathbf{x}) - p\_{X,Y}(\cdot|\mathbf{x})$ is generally unbounded, but its VE, $\tilde{p}\_r(\cdot|\mathbf{x}') - \tilde{p}\_{X, Y}(\cdot|\mathbf{x}')$ is bounded by $||\tilde{q}||\_\infty$, making it possible to obtain a PAC bound like **Theorem 2** (Please refer to the proofs for the more details).
>
> > The idea to use an auxiliary distribution is similar to importance sampling, are they related somehow?
>
> The "auxiliary distribution" in importance sampling is the distribution function chosen similar to the target distribution to be sampled, whereas auxiliary distribution in our manuscript rather means the mapping between label space and auxiliary label space. Despite sharing the name "auxiliary distribution", we do not consider these two concepts to have anything in common.

---

### Official Review · Reviewer_ushT · 2023-10-30

**Soundness:** 2 fair
**Presentation:** 2 fair
**Contribution:** 2 fair
**Rating:** 3
**Confidence:** 4

**Summary:**

The paper deals with the problem of sampling from the conditional distribution of $Y$ given $X=x$, based on a sample of size $N$ denoted by $(X_1,Y_1),\ldots, (X_N,Y_N)$. The Vicinal Estimation method is introduced that, roughly speaking, consists in replacing the original problem by that of estimating the conditional distribution of $Y$ given $X'=x'$, where $X'$ is a random label artificially generated from a conditional distribution $q(.|X = x)$. The main contribution of the paper is theoretical: it provides an upper bound on the distance between the learned conditional density and the true conditional density. Taking the set of discriminators to be equal to a subset in the Barron space, it is shown that it is possible to get an error that behaves itself as $N^{-1/(6+2d_X)}$, where $d_X$ is the dimension of the set of the labels $X$.

**Strengths:**

1. Studies an important problem.
2. Obtains a new bound on the error of conditional sampling

**Weaknesses:**

1. Section 4, which is one of the most important ones is poorly written. The definition of the generator that is analyzed is not given in full detail. In particular, neither the set of the discriminators nor the set of the generators involved in the min-max problem of the GAN are clearly specified. In addition, it is not clear in Eq 19 what is understood by $\tilde p_{X,Y}(y|x')$.
2. Section 5 is very dense and hard to read. The results are quite technical so it is impossible to say whether they are plausible or not without checking the proofs line by line. Given the limited time I had to review the paper, I could not check the proofs provided in the supplementary material. I believe that it would be much better for such a paper to be submitted to a journal, where more space may be used for stating the main theorems and providing some explanations and intuitions, as well as more time could be left to the reviewers for checking the details of the proofs.
3. It seems that the results of the paper do not imply that the vicinal estimation method is better than the vanilla cGAN. Indeed, the fact that the rate of convergence does not depend on the dimension of Y might very well be a consequence of the choice of the set of discriminators. In particular, it is well known that the lower bound (1) is due to the fact that the set of discriminators defining the $W_1$ (which is a lower bound on $W_2$) -- the set of 1-Lipschitz functions -- is too large.  Replacing this set by an RKHS with a bounded kernel leads to the dimension independent rate $N^{-1/2}$.

**Questions:**

See above.

---

> ### Author Response · Authors · 2023-11-16
> **Response to Reviewer ushT**
>
> We appreciate the reviewer **ushT** for the detailed comments and hope to address all of the questions and concerns below.
>
> > Section 4, which is one of the most important ones is poorly written. The definition of the generator that is analyzed is not given in full detail. In particular, neither the set of the discriminators nor the set of the generators involved in the min-max problem of the GAN are clearly specified. In addition, it is not clear in Eq 19 what is understood by $\tilde{p}\_{X,Y}(y|x')$.
>
> The details of the GANs used in the proposed framework are described in Section 4. Specifically, they are as follows:
>
> - Set of the discriminator : $\tilde{D}\_a$ defined by Equation (14) parameterized by $a$
> - Set of the generator : For an arbitrary generator $G$, we use **Algorithm 2** to sample from $\tilde{p}\_g(\cdot|\mathbf{x}')$. So the set of generators is not limited.
> - $\tilde{p}\_{X,Y}(y|x')$ is a conditional distribution, which is induced from the joint distribution $\tilde{p}\_{X,Y}$ which is a VE of the empirical distribution. We have added the desciption for $\tilde{p}\_{X,Y}$ in the revised manuscript.
>
> > Section 5 is very dense and hard to read. The results are quite technical so it is impossible to say whether they are plausible or not without checking the proofs line by line. Given the limited time I had to review the paper, I could not check the proofs provided in the supplementary material. I believe that it would be much better for such a paper to be submitted to a journal, where more space may be used for stating the main theorems and providing some explanations and intuitions, as well as more time could be left to the reviewers for checking the details of the proofs.
>
> To the best of the authors' knowledge, there are many theoretic papers in ICLR, including generalization error analysis which our paper focuses on. While the journal will be more suitable for the papers with complex mathematical proofs, we believe that our manuscript does not contain such complex proofs as the proofs are not extremely long (3 pages in appendices) and do not involve extremely complex mathematical concepts. Our proposed model is validated by the experimental results which also support the theoretical analysis.
>
> > It seems that the results of the paper do not imply that the vicinal estimation method is better than the vanilla cGAN. Indeed, the fact that the rate of convergence does not depend on the dimension of Y might very well be a consequence of the choice of the set of discriminators. In particular, it is well known that the lower bound (1) is due to the fact that the set of discriminators defining the $W\_1$ (which is a lower bound on $W\_2$
> ) -- the set of 1-Lipschitz functions -- is too large. Replacing this set by an RKHS with a bounded kernel leads to the dimension independent rate $N^{-1/2}$.
>
> First of all, we are not aware of any results showing that the generalization error in $W\_1$ can be bounded to $N^{-1/2}$ by RKHS kernel regularization, as you mention in the last sentence. To the best of the authors' knowledge, Bias Potential Model (2021) was almost the first to achieve dimension-independent generalization error for unconditional GANs, except when bounding $p\_g$ itself, and the best we know of is $N^{-1/4}$ for KL divergence and $N^{-1/6}$ for $W\_2$.
>
> Even if dimension-independent generalization error can be easily achieved for unconditional GANs as mentioned, it cannot be extended to conditional GANs (vanilla cGANs?) due to a problem named "Lack of Conditional Sample" : even if a generalization error is obtained for each fixed label value, the conditional samples of that label value will be much smaller than the total number of samples.
>
> One of the key contributions of our work is to introduce the concept of vicinal estimation, which allows us to obtain a sufficient number of conditional samples for arbitrary (auxiliary) label values.

---

> > ### Comment · Reviewer_ushT · 2023-11-22
> > **Thank you for the response**
> >
> > I would like to thank the authors. However, I still find that the paper is poorly written and needs a substantial revision. In the current version, the lack of clarity makes it hard to assess the significance and the correctness of the mathematical results.
> >
> > I repeat the example I gave in my review. The authors write "$\tilde D$ is trained to maximize $\hat V'(G,\tilde D)$ and $G$ is trained to minimize $\hat V(G,\tilde D)$. This is the last sentence of Section 4. Nothing in this sentence, nor in the text before, informs the reader over which sets the mentioned maximization and minimization are conducted. The reply above does not answer this question. The authors say that the discriminators are parameterized by $a$, without making it precise which set this function belongs to. If $a$ is the parameter, does it mean that one maximizes $\hat V'$ with respect to $a$? My initial understanding was that the parameters are $w$ and $b$, and the maximization is wrt these parameters. Similarly, the answer above concerning the set over which the minimization wrt to $G$ is conducted is puzzling.
> >
> > Finally, the conditional density $\tilde p$ corresponding to the VE is defined in the paper by Eq (9). This equation is valid for joint distributions that admit a density $p$ wrt to the Lebesgue measure. The empirical distribution $p_{X,Y}$ is not absolutely continuous wrt the Lebesgue measure. Therefore, the corresponding VE should be defined without relying on densities. Of course, I am not saying that this is not possible. I am just pointing to the fact that it is not done in the paper.

---

### Official Review · Reviewer_JqUD · 2023-10-30

**Soundness:** 2 fair
**Presentation:** 2 fair
**Contribution:** 3 good
**Rating:** 5
**Confidence:** 4

**Summary:**

This article studies the generalization property of conditional GANs to estimate conditional probabilities p(y|x). By adding conditional data samples with Vicinal estimation and using 1-layer neural network discriminator, the proposed model is able to achieve a Wasserstein error which does not grow with the dimension of y. As long as the dimension of of x is small, it overcomes the curse-of-dimensionality when the dimension of y is large.

**Strengths:**

This article makes a creative combination of 2 existing ideas: one based on 1-layer neural network discriminator based on Barron space and early stopping in learning to achieve dimension-free generalization. The other idea is based on vicinal estimation to define a new training loss to increase effective number of samples.

**Weaknesses:**

The theoretical results are somehow not very clear, making it hard and inconsistent to understand how the whole idea is connected to the Wasserstein-2 error used in the evaluation of cGANs. I would suggest a clearer explanation in the rebuttal phase.

**Questions:**

-	What is the functional space of the discriminator D(y,x) in eq. 5, and how is it related to the W-1 distance in eq. 6? I suppose you are assuming the (y,x) -> D(y,x) is Lipschitz, but for the W-1 distance, we only need y -> D(y,x) to be Lipschitz for any x.
-	The tilde D_alpha(y,x’) defined in eq 14, is it also Lipschitz? I do not see why this is so (even based on eq. 22) and it seems to be inconsistent to your previous definition.
-	What is the definition of tilde p_{x,y} and p(x’) in eq. 20? Is p(x’) an empirical distribution as in eq. 7? It seems not to be the case according to theorem 2 and 3. Then I am confused of the over-all setting if you are the true distribution p(x’) of x’ in your training loss.
-	Do you need to assume that the kernel is characteristic in Theorem 3 ? It is not clear what are the above conditions in Theorem 5. Please make it clearer. Could you also explain why there is no d_y in the statement of Theorem 5?

Minor:

-	Why did you introduce a RKHS regulation in the loss V’ in eq. 18? What would happen if there is no regulation?
-	What is y used for in eq. 25 ? I do not see it to appear on the right hand side.
-	How do you get the optimal scale of ||tilde q||_inf after eq. 33 ?

---

> ### Author Response · Authors · 2023-11-16
> **Response to Reviewer JqUD**
>
> We appreciate the reviewer **JqUD** for the detailed comments and hope to address all of the questions and concerns below.
>
> > What is the functional space of the discriminator D(y,x) in eq. 5, and how is it related to the W-1 distance in eq. 6? I suppose you are assuming the (y,x) -> D(y,x) is Lipschitz, but for the W-1 distance, we only need y -> D(y,x) to be Lipschitz for any x.
>
> As mentioned, it is enough for $D$ to be Lipschitz continuous with respect to $\mathbf{y}$. We have corrected the sentence.
>
>
> > The tilde D_alpha(y,x’) defined in eq 14, is it also Lipschitz? I do not see why this is so (even based on eq. 22) and it seems to be inconsistent to your previous definition.
>
> In the definition of Equation (14), $D_a(\mathbf{y},\mathbf{x})$ is not Lipschitz. In our proposed model, we do not need the Lipschitzness of $D$ because we are applying RKHS norm regularization rather than enforcing Lipschitz like Wasserstein cGAN.
>
>
> > What is the definition of tilde p_{x,y} and p(x’) in eq. 20? Is p(x’) an empirical distribution as in eq. 7? It seems not to be the case according to theorem 2 and 3. Then I am confused of the over-all setting if you are the true distribution p(x’) of x’ in your training loss.
>
> For $\tilde{p}\_{X,Y}$ , it is a vicinal estimate of the empirical distribution defined in Equation (2), which can be obtained by substituting $\tilde{p}\_{X,Y}$ for $p$ in Equation (8). We clarify it before Equation (19) in the revised manuscript to indicate that $\tilde{p}\_{X,Y}$ is the VE of the empirical distribution.
>
> What you refer to as $p(\mathbf{x}')$ in Equation (20) is $\rho(\mathbf{x}')$, which means a random distribution, not the actual distribution of auxiliary label values. Once $\rho(\mathbf{x}')$ is known, we can obtain a generalization bound. Moreover, we can consider various $\rho(\mathbf{x}')$ to our main results such as **Corollary 2**. For example, as mentioned in the manuscript, when $\tilde{q}$ is a perturbation, we can substitute $p_r(\mathbf{x})$ into $\rho$ to get a generalization error between $p_g$ and $p_r$.
>
>
> > Do you need to assume that the kernel is characteristic in Theorem 3 ? It is not clear what are the above conditions in Theorem 5. Please make it clearer. Could you also explain why there is no d_y in the statement of Theorem 5?
>
> In **Theorem 3**, it is not necessary for the kernel to be characteristic. As mentioned, the various conditions assumed in **Theorem 5** seem to be scattered, so we have summarized the exact conditions in **Theorem 5**.
>
> Finally, the reason why $d_\mathbf{y}$ does not appear in **Theorem 5** and **Corollary 2** is because of the Big-O notation to emphasize the decay rate for a dataset size $N$. If we write the right-hand side correctly, it is multiplied by a sublinear term for $d_\mathbf{y}$, which is omitted in Big-O Notation because it is a term independent of $N$.
>
>
> > Why did you introduce a RKHS regulation in the loss V’ in eq. 18? What would happen if there is no regulation?
>
> The RKHS norm regularization introduced in Equation (18) and elsewhere is a variation of prior work (Weinen & E, 2021) that introduced a dimension-free generalization error in unconditional GANs. If there is no such regularization, i.e. if the error had the form of $V'(G,\tilde{D}) = \mathbb{E}\_{\mathbf{x}' \sim \rho(\mathbf{x}')} \left[ \mathbb{E}\_{\mathbf{y} \sim \tilde{p}\_r(\mathbf{y}|\mathbf{x}')}[\tilde{D}(\mathbf{y},\mathbf{x}')] - \mathbb{E}\_{\mathbf{y} \sim \tilde{p}\_g(\mathbf{y}|\mathbf{x}')}[\tilde{D}(\mathbf{y},\mathbf{x}')] \right]$, it is possible to convert it to a problem of minimizing $\mathbb{E}\_{\mathbf{x}' \sim \rho(\mathbf{x}')} W\_1 (\tilde{p}\_g(\cdot|\mathbf{x}'), \tilde{p}\_r(\cdot|\mathbf{x}'))$ by assuming Lipschitzness on \tilde{D}. In this case, we're still facing the curse of dimensionality and cannot get a reasonable generalization bound.
>
> > What is y used for in eq. 25 ? I do not see it to appear on the right hand side.
>
> Note that $\mathbf{y}$ is used as the first argument of the kernel in Equation (25). We've omitted it because it's a notation commonly used in convolution, but it's technically equivalent to $\int k(\mathbf{y},\mathbf{y}') d(\tilde{p}_r(\mathbf{y}'|\mathbf{x}') - \tilde{p}_g^t (\mathbf{y}'|\mathbf{x}'))(\mathbf{y}')$.
>
> > How do you get the optimal scale of ||tilde q||_inf after eq. 33 ?
>
> If we leave $N$ and $Q=||\tilde{q}||\_\infty$ to find $||\tilde{q}||\_\infty$ that minimizes the right-hand side of Equation (33), it becomes equal to $AN^{-1/6}Q^{1/3} + BQ^{-1/d_{\mathbf{x}}}$. The $Q$ that minimizes this expression is easily obtained via differentiation.

---

> > ### Comment · Reviewer_JqUD · 2023-11-22
> > **Thanks for addressing my questions**
> >
> > I shall still maintain my score. A remark to improve the article is to study the case where you do not use the rho(x) in your training loss of cGANs, as this can be the true data distribution p_r (x), rather than the empirical distribution.

---

### Official Review · Reviewer_GnZt · 2023-11-06

**Soundness:** 3 good
**Presentation:** 2 fair
**Contribution:** 2 fair
**Rating:** 5
**Confidence:** 2

**Summary:**

This paper studies the problem of learning a conditional distribution with a continuous input and potentially high-dimensional output, using a variant of a generative adversarial network (GAN) with a "vicinal estimation" mechanism to address the sparsity of observed labels in the label-space. The paper first provides theoretical guarantees showing that the proposed approach (together with an early stopping mechanism) learns the true conditional distribution in $2$-Wasserstein distance and then provides a toy experiment in which the proposed method is demonstrated to learn a synthetic conditional distribution more efficiently than a baseline cGAN.

**Strengths:**

The vicinal estimation (VE) approach seems novel in the setting of conditional generation. The paper presents precise theoretical guarantees supporting the proposed method. The synthetic data experiment is clearly explained and its results are quite precise and illustrative.

**Weaknesses:**

1. I suggest removing "and Early Stopping" from the title of the paper, as (a) this is hardly discussed in the paper and (b) it seems to be a direct adaptation of Yang & E (2021) to the conditional setting rather than a really novel contribution.

2. Eq. (1): In this sentence, $p$ should be quantified more carefully; there are obvious counterexamples (e.g, when $p$ is a single point mass) for which this statement, as written, is false.

3. I found it counterintuitive that $X$ is used for the label and $Y$ is used for the generated image. While this makes sense in terms of the inputs and outputs of a cGAN, it conflicts with much more common settings in the ML literature (e.g., image classification, where $X$ is the image and $Y$ is the label), causing me to be confused through much of the paper until I went back and re-read the beginning of Section 3. I don't necessarily suggest changing this (especially if it is consistent with other paper on conditional generative modeling), but perhaps it is worth adding a sentence to explicitly point out this possible point of confusion.

4. Section 4: The notation in here seemed unnecessarily complicated. Isn't $q_x$ simply the conditional distribution of $X'$ given $X$. Why introduce a new notation rather than simply writing it as such? If I understand correctly, later parts could be written much more readably (e.g., in Lemma 1, $m_{\tilde q}(x') = \mathbb{E}[X|X']$ and $d_{\tilde q}(x') = \sqrt{\mathbb{E}[||X - \mathbb{E}[X|X']||^2|X']}$, etc.).

5. Just before Eq. (11), a minor wording suggestion: "a VE vanishes the conditional information" -> "a VE reduces the conditional information"

6. Theorem 1: Isn't $||\tilde q^L||_\infty = L^{-d_x} ||\tilde q^1||_\infty$? If so, perhaps explanding this would make the statement it a bit more intuitive.

7. Although the work is intended to be theoretical, it would be strengthened by any discussion of real-world relevance (e.g., describing some applications where the proposed vicinal estimation approach might be useful). Experiments on real-world data would further strengthen the paper.

**Questions:**

1. Just before Eq. (11)
> uniform auxiliary distributions on $\mathcal{X}'$ erase completely the information of labels in the data

If I understand correctly, what really matters here is how much information about $X$ is preserved by $X'$, rather than the particular distribution of $X'$. For example, doesn't any distribution of $X'$ that is independent of $X$ completely the information of labels? Or is there some reason the uniform distribution is special here?

2. Just after Lemma 1
> Note that the quantities $m_{\tilde q}(x′)$ and $d_{\tilde q}(x′)$ only depend on the auxiliary distribution $q$, not on the distribution $p$.

I found this sentence confusing, as $m_{\tilde q}(x′)$ and $d_{\tilde q}(x′)$ depend on the inverse auxiliary distribution $\tilde q$, which is related to the auxiliary distribution $q$ in a way that depends on $p$ (namely, through Eq. (10)). Perhaps this was a typo (i.e., "auxiliary distribution $q$" should have been "inverse auxiliary distribution $\tilde q$")? Even so, I don't really understand the point of this sentence.

---

> ### Author Response · Authors · 2023-11-16
> **Response to Reviewer GnZt (Part 1)**
>
> We appreciate the reviewer **GnZt** for the detailed comments and hope to address all of the questions and concerns below.
>
> > I suggest removing "and Early Stopping" from the title of the paper, as (a) this is hardly discussed in the paper and (b) it seems to be a direct adaptation of Yang & E (2021) to the conditional setting rather than a really novel contribution.
>
> We have removed "and Early Stopping" from the title in the revised manuscript. The OpenReview post title is not currently editable, but we will edit it when it is.
>
> > Eq. (1): In this sentence, $p$ should be quantified more carefully; there are obvious counterexamples (e.g, when $p$ is a single point mass) for which this statement, as written, is false.
>
> Equation (1) requires the condition that $p$ is absolutely continuous with respect to the Lebesgue measure. We have added this condition after Equation (1).
>
>
> > I found it counterintuitive that $X$ is used for the label and $Y$ is used for the generated image. While this makes sense in terms of the inputs and outputs of a cGAN, it conflicts with much more common settings in the ML literature (e.g., image classification, where is $X$ the image and $Y$ is the label), causing me to be confused through much of the paper until I went back and re-read the beginning of Section 3. I don't necessarily suggest changing this (especially if it is consistent with other paper on conditional generative modeling), but perhaps it is worth adding a sentence to explicitly point out this possible point of confusion.
>
> In this study, we followed the notation and considered an error analysis from the perspective of Conditional Density Estimation. For this purpose, we used $\mathbf{x}$ to denote a label and $\mathbf{y}$ to denote an image, as used in the studies related to Conditional Density Estimation. We have added a sentence at the beginning of Section 3 to clarify that $\mathbf{x}$ is a label and $\mathbf{y}$ is an image.
>
>
> > Section 4: The notation in here seemed unnecessarily complicated. Isn't $q_{x}$ simply the conditional distribution of $X'$ given $X$. Why introduce a new notation rather than simply writing it as such? If I understand correctly, later parts could be written much more readably (e.g., in Lemma 1, $m_q(x') = \mathbb{E}[X|X']$ and $d_q(x') = \sqrt{\mathbb{E}[||X-\mathbb{E}[X|X']||^2|X']}$ , etc.).
>
> The reason for defining the value of $q$ or $\tilde{q}$ is that we need the value of an explicit probability density function. For example, in our analysis, $||\tilde{q}||_{\infty}$ is an important factor in the generalization bound, and the definition of this value requires the definition of the probability density function $\tilde{q}$.
>
>
> > Just before Eq. (11), a minor wording suggestion: "a VE vanishes the conditional information" -> "a VE reduces the conditional information"
>
> We have modified the sentence.
>
> > Theorem 1: Isn't $||\tilde q^L||_\infty = L^{-d_x} ||\tilde q^1||_\infty$? If so, perhaps explanding this would make the statement it a bit more intuitive.
>
> Just before **Theorem 1**, we added a sentence to emphasize the fact that $d_{\tilde{q}^L}(\textbf{x}') \propto L^{-1}$ and $||\tilde{q}^L||_{\infty} \propto L^{-d_{\textbf{x}}}$.
>
>
> > Although the work is intended to be theoretical, it would be strengthened by any discussion of real-world relevance (e.g., describing some applications where the proposed vicinal estimation approach might be useful). Experiments on real-world data would further strengthen the paper.
>
> In Conditional cGAN (CcGAN), Ding et al. (2023) utilizes the concept of perturbing the label value in one-dimensional continuous labels and demonstrates the effectiveness of their proposed method for real-world conditional image generation tasks. As perturbation is a special example of vicinal estimation proposed in our manuscript, we can expect that VE and our proposed method can also improve the performance of cGAN models.
>
> There is a technical issue to undertake a supplementary experiment for real-world data, as we require a more intricate discriminator model based on the Barron space. However, we are currently constructing a model to conduct experiments on a real-world image dataset. We will report the results if we obtain them before the rebuttal period.

---

> ### Author Response · Authors · 2023-11-16
> **Response to Reviewer GnZt (Part 2)**
>
> > Just before Eq. (11)
> >   > uniform auxiliary distributions on $\mathcal{X}'$ erase completely the information of labels in the data
> >
> > If I understand correctly, what really matters here is how much information about $X$ is preserved by $X'$, rather than the particular distribution of $X'$. For example, doesn't any distribution of $X'$ that is independent of $X$ completely the information of labels? Or is there some reason the uniform distribution is special here?
>
> We use the uniform auxiliary distribution as an extreme example where the information of $X$ is not preserved in $X'$.
>
>
> > Just after Lemma 1
> >   > Note that the quantities $m_{\tilde{q}}(x')$ and $d_{\tilde{q}}(x')$ only depend on the auxiliary distribution $q$, not on the distribution $p$.
> >
> > I found this sentence confusing, as $m_{\tilde{q}}(x')$ and $d_{\tilde{q}}(x')$ depend on the inverse auxiliary distribution $\tilde{q}$, which is related to the auxiliary distribution in a way that depends on $p$ (namely, through Eq. (10)). Perhaps this was a typo (i.e., "auxiliary distribution" should have been "inverse auxiliary distribution ")? Even so, I don't really understand the point of this sentence.
>
> We have modified "auxiliary distribution $q$" to "inverse auxiliary distribution $\tilde{q}$". The reason we emphasize this fact is that $\tilde{q}$ is the most central function in Vicinal Estimation, as only the condition on $\tilde{q}$ is needed to obtain these values and the Generalization Bound that follows. Moreover, only $\tilde{q}$ is used in actual implementation in **Algorithm 2**. We've modified the sentence after **Lemma 1** to emphasize this fact.

---

### Author Response · Authors · 2023-11-16
**Summary of Updates to the Manuscript**

We appreciate the reviewers for the detailed comments. We describe the changes we made to the manuscript to incorporate the reviewers' feedback.

- We have removed "and Early Stopping" from the manuscript title. (**GnZt**)
- After Equation (1), we have added the condition that $\mu$ must be absolutely continuous with respect to the Lebesgue measure. (**GnZt**)
- We have added a sentence at the beginning of Section 3 to clarify that $\mathbf{x}$ is a label and $\mathbf{y}$ is an image. (**GnZt**)
- Minor word modification before Equation (11) : "a VE vanishes the conditional information" -> "a VE reduces the conditional information" (**GnZt**)
- Just before **Theorem 1**, we have added a sentence to emphasize the fact that $d_{\tilde{q}^L}(\textbf{x}') \propto L^{-1}$ and $||\tilde{q}||\_{\infty} \propto L^{-d_{\textbf{x}}}$. (**GnZt**)
- Just after **Lemma 1**, we have modified "auxiliary distribution $q$" to "inverse auxiliary distribution $\tilde{q}$". (**GnZt**)
- Just after **Lemma 1**, we have replaced the last sentence to emphasize the important of $\tilde{q}$. (**GnZt**)
- After Equation (4), we have modified the condition of $D$. (**JqUD**)
- Just before Equation (19), we have added a description ("VE of empirical distribution") of $\tilde{p}_{X,Y}$. (**JqUD**, **ushT**)
- We have provided a clear sentence on the assumptions for Theorem 5. (**JqUD**)
- Other minor typos are corrected.

---

### Meta-Review · Area_Chair_YAfC · 2023-12-07

**Metareview:**

All the reviewers find it difficult to understand important parts of the paper. I recommend that the authors go through an extra round of rewriting and submission process.

**Justification For Why Not Higher Score:**

If the paper were better written, I expect it to be acceptable to a leading machine learning venue.

**Justification For Why Not Lower Score:**

n/a

---

### Decision · Program_Chairs · 2024-01-16

Reject